# URB - Urban Routing Benchmark for RL-equipped Connected Autonomous Vehicles

**Ahmet Onur Akman**[1,*]**, Anastasia Psarou**[1]**, Michał Hoffmann**[2]**, Łukasz Gorczyca**[2]**,
Łukasz Kowalski**[3]**, Paweł Gora**[2]**, Grzegorz Jamróz**[2]**, Rafał Kucharski**[2]

[1] Doctoral School of Exact and Natural Sciences, Jagiellonian University, Kraków, Poland
[2] Faculty of Mathematics and Computer Science, Jagiellonian University, Kraków, Poland
[3] Urban Policy Observatory, Institute of Urban and Regional Development, Warsaw, Poland

## Abstract

Connected Autonomous Vehicles (CAVs) promise to reduce congestion in future urban networks, potentially by optimizing their routing decisions. Unlike for human drivers, these decisions can be made with collective, data-driven policies, developed using machine learning algorithms. Reinforcement learning (RL) can facilitate the development of such collective routing strategies, yet standardized and realistic benchmarks are missing. To that end, we present URB: Urban Routing Benchmark for RL-equipped Connected Autonomous Vehicles. URB is a comprehensive benchmarking environment that unifies evaluation across 29 real-world traffic networks paired with realistic demand patterns. URB comes with a catalog of predefined tasks, multi-agent RL (MARL) algorithm implementations, three baseline methods, domain-specific performance metrics, and a modular configuration scheme. Our results show that, despite the lengthy and costly training, state-of-the-art MARL algorithms rarely outperformed humans. The experimental results reported in this paper initiate the first leaderboard for MARL in large-scale urban routing optimization. They reveal that current approaches struggle to scale, emphasizing the urgent need for advancements in this domain.

## 1 Introduction

Recent technological [57] and algorithmic [35] advancements let us believe that in the foreseeable future, Connected Autonomous Vehicles (CAVs) will enter our cities and start driving alongside human drivers [44, 68]. These vehicles will not only successfully navigate through the traffic complexity and arrive safely at the destination, but also make independent routing decisions: which route to select to reach the destination. A possible future scenario could be as follows:

*In the small French town of St. Arnoult, 40% of drivers decide to switch to autonomous driving mode, delegating their routing decisions. Then, either each vehicle or the central controller applies some algorithm to select routes to minimize travel costs.*

This raises a series of significant and open research questions:

1. Which algorithm is most suitable for collective urban fleet routing?
2. Does RL outperform alternative operations research (OR) or machine learning (ML) methods? If yes, how efficient can the training be, what reward formulation best captures the objective, should the solution be centralized or decentralized, how can we formulate useful observations within practical constraints? How detailed environment simulations are needed?
3. How does the problem scale with network complexity, number of agents, and planning horizon?

---

*Corresponding author: Ahmet Onur Akman, `onur.akman@uj.edu.pl`

39th Conference on Neural Information Processing Systems (NeurIPS 2025) Track on Datasets and Benchmarks.

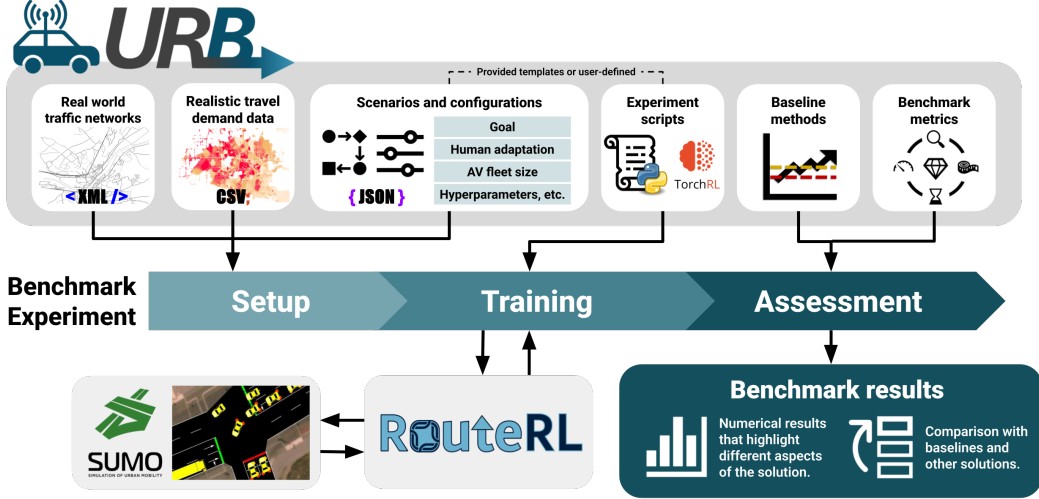

Figure 1: URB is a comprehensive benchmarking framework for MARL methods in solving CAV routing tasks in a mixed urban traffic environment. It enables end-to-end assessment through a collection of 29 real-world traffic networks, realistic demand patterns, baseline methods, domain-specific performance indicators, and a flexible parameterization scheme.

4. What is the impact of applying such algorithms on: the transportation system, CAVs, and non-CAV drivers? By focusing only on algorithmic goals, are we overlooking any significant aspects of CAV deployment that may be detrimental to our urban societies (increased emissions, mileage, variability, inequality, etc.)?

Those questions are timely, yet open [52, 33]. Therefore, in this study, we introduce URB - a comprehensive benchmarking framework for MARL methods in solving CAV routing tasks in a mixed (CAV-human) urban traffic environment. We outline its motivation, theoretical foundation, design, and key features. We then initiate the first URB leaderboard with four of the most established MARL algorithms on a representative scenario and document how each solution affects different system efficiency statistics. Finally, we discuss URB's practical potential and limitations, and outline the future directions we intend to pursue.

The human-CAV urban routing problem is interdisciplinary, intersecting traffic engineering (with detailed vehicle dynamics and traffic flow properties), transport engineering (with day-to-day route choice behavior in human daily demand patterns), and machine learning (with a discrete optimization problem on a huge action space in a non-stationary environment). URB bridges experts from the above fields, allowing them to contribute without in-depth domain expertise. In particular, thanks to SUMO [40] integration and interface of the common human route choice models and demand patterns, machine learning researchers can test their custom algorithms on realistic traffic scenarios. Likewise, transport researchers, thanks to the provided TorchRL [9] implementations, may use state-of-the-art RL algorithms and test their impact on custom traffic and travel demand scenarios.

Covering all practical scenarios of future urban routing is a challenging task. URB is a novel benchmarking framework that effectively addresses this challenge through its flexibility and extensibility. Compared to prominent benchmarks in MARL (as detailed in Table 6), URB delivers (i) task diversity, (ii) high scenario coverage, (iii) experimentation customizability, and (iv) high component extensibility (custom tasks, traffic networks, demand, algorithms, etc.). As a result, URB stands as not only a valuable tool for catalyzing advancements in its domain, but also for inspiring realistic and high-coverage benchmarks.

With URB, we aim to trigger positive competition within the RL community to propose new algorithms, and test them on a diverse set of tasks and instances. Hopefully, URB leaderboard will eventually be dominated by efficient, scalable, reliable, and socially aware solutions that, when deployed in real networks, will improve the performance of all the parties involved. URB will be gradually extended with problems that will arise in the future, when CAVs will start operating in our cities in various configurations, with candidate solutions (from within the RL community and outside) evaluated

on a wide set of measures and problem instances. This is particularly timely before CAV fleets are deployed, to inform society about potential threats and motivate vehicle developers to propose sustainable fleet operating algorithms.

URB's codebase[2] and data collection[3] are publicly available in online repositories under non-restrictive licenses, as detailed in Appendix C.1.

## 1.1 Urban routing and (MA)RL - significance and challenges calling for benchmark

The urban routing problem with CAVs is not only *significant* for future societies, but also *challenging*. The CAVs promise to relieve our congested networks by enabling a new class of routing behaviors, potentially making better use of scarce resources (capacity) for better global and user-level performance. This translates into minutes saved daily and hours saved annually for all commuters, and tons of fuel and $CO_2$ emissions saved at the system level.

Realizing this potential, however, requires solving a highly non-trivial problem. With thousands of agents (up to millions in megacities) taking simultaneous actions in large action spaces (here we sample only a few routes, while the choice set grows exponentially with network size), the environment is not only non-deterministic (due to the stochastic nature of traffic and the heterogeneity of people), but also non-stationary (as agents compete for limited resources in a game-like fashion) and costly to simulate. One environment run can take up to hours (with detailed simulators like SUMO) in real-size cities, while MARL algorithms often require millions of training episodes. At the same time, standard discrete OR techniques fail in such large action spaces [43].

Existing studies applied MARL to tackle the urban routing problem. In [63], drivers use MARL with different rewards to minimize congestion, but consider only homogeneous agent populations and global reward; in [53], authors proposed a regret-minimization for MARL route selection with external traffic data, and [61] used a centralized controller with a simplified macroscopic traffic simulation. Finally, [3] introduced RouteRL, a simulation framework for mixed urban route choice, allowing for experimentation with custom AV deployment scenarios in human systems. With URB, we build on it and extend, presenting a benchmarking environment that features a variety of tasks and realistic traffic scenarios, introduces baselines and novel evaluation metrics, and simulates with a microscopic traffic simulator, SUMO, thanks to its RouteRL integration (see Appendix E.1 for details).

In line with the aforementioned studies, we argue that solving the urban routing problem with RL may be the most intuitive direction, as it can be naturally reframed as a decision-making task, and the optimization target can be intuitively formulated as a reward signal. The human day-to-day route-choice learning process [69] naturally resembles the classical RL training loop. RL facilitates experience-driven learning and adaptability, which are particularly useful for complex, large-scale, and dynamic problems. Moreover, existing RL research introduced approaches that could potentially address the challenges involved in the urban route choice problems, such as managing a large group of agents [12, 10], communication mechanisms [71, 64], and developing network-agnostic routing strategies [6, 46]. Moreover, multi-agent learning may be an effective way to decompose this large-scale optimization problem into smaller learning tasks.

Nonetheless, as our results suggest, the current frontiers of MARL algorithms fall short of addressing the problem complexities. To this end, we introduce a unified ground to stimulate advancements in this domain, similarly to other successful traffic-related RL benchmarks like FLOW [34], RESCO [5], or [65]. We are optimistic that the RL community, challenged by URB will propose reliable, generalizable, and efficient solutions. To the best of our knowledge, no existing benchmark or RL environment combines all the mentioned characteristics into a single problem instance (as we detail in Appendix D); thus, solving it will support the general MARL development and benefit any other domain where MARL is applicable.

---

[2]`https://github.com/COeXISTENCE-PROJECT/URB`
[3]`https://doi.org/10.34740/kaggle/ds/7406751`

## 2 Background

### 2.1 Urban routing problem with human drivers

We consider a specific variation of the classic traffic assignment problem (TAP) [13] (known in game theory as a repeated congestion game [22]) to represent the actual commuting decisions made by drivers in congested urban traffic networks every day worldwide. Each agent (driver) selects the subjectively optimal route to reach their destination at minimal cost [47]. Agents' decisions, due to limited capacity, contribute to system travel costs (congestion). The demand (agents with their origins, destinations, and departure times) is fixed. Every day (later formalized as an RL episode), they update their knowledge with the recent experience and select the subjectively optimal routes [72]. Such a system is expected, after sufficient learning, to reach the so-called User Equilibrium [42], a specific version of Nash Equilibrium [67], where none of the drivers may improve payoffs (here, simply the travel time) by unilaterally switching routes. This, however, relies on very strict conditions (perfect rationality and knowledge about the traffic conditions [11]) that are not met in practice, so real-world conditions (as well as microscopic agent-based models resembling them) only approximate the equilibrium conditions.

In URB, the realistic microscopic traffic simulator SUMO is used to reproduce travel costs (here, simply the travel times) and a generic human learning model (HLM) is used to update knowledge, on which each human driver makes subjectively optimal routing decisions [20]. In brief, every day $\tau$, each human agent $i$ executes an action $a_{i,\tau}$ from the set of routes $K_i$, to maximize own expected utility $U$ (reward, here simply the negative travel time) $a_{i,\tau} = \arg\max_{k \in K_i} U_{i,\tau,k}$ (where $K_i$ is the set of routes considered by agent $i$). Daily, the agent $i$ updates own expectations with recent experienced travel time $t_{i,\tau,k}$:   $U_{i,\tau,k} = HLM(U_{i,\tau-1,k}, t_{i,\tau,k})$. Details for HLM are provided in Appendix E.3. In this research, we use a standard yet stable and reproducible model to isolate the impact of algorithms from non-deterministic, heterogeneous, complex, and suboptimal travel behavior.

### 2.2 Urban routing problem with autonomous vehicles

While the above problem has been present since the first traffic jams in our cities and is well studied, its new formulation [3], where limited resources of urban traffic networks are shared between CAVs and human-driven vehicles, is less known. When autonomous vehicles, rather than humans, decide how to reach a destination, the problem changes significantly: (i) instead of making behavioral routing decisions, CAVs seek optima algorithmically, (ii) instead of subjectively perceived costs, CAVs have predefined reward functions, (iii) which, unlike for humans, can go beyond their own travel time and (iv) can become shared and potentially collective.

### 2.3 Problem formulation

During several consecutive episodes (days of commute $\tau$), each agent (vehicle $i$) makes a pre-trip routing decision $a_i^\tau$, i.e., chooses a path from the precomputed set of routes. Humans $i \in \mathcal{N}_{HDV}$ use the behavioral model explained above, while CAVs $j \in \mathcal{N}_{CAV}$ use their routing policies $\pi$ to select an action $a$ in a given (observed) state $o$: $a \sim \pi(a|o)$. Agent rewards (from travel times yielded by SUMO) depend on the joint actions of all agents. In URB, we typically consider scenarios where: (1) humans train first and stabilize their action probabilities, followed by (2) the mutation where some agents become CAVs and (3) CAVs start training by applying some learning algorithm to maximize a reward signal, and finally (4) in the testing phase, CAVs do not learn and roll out trained policies for several days (episodes).

Building on [3], we formalize this problem as the Agent-Environment Cycle (AEC) game [60], where in each episode, agents $\mathcal{N}$, either human or CAV, in order $v$ of departure time, select a route from their action space. Within the episode, the system evolves according to traffic dynamics ($P$ simulated with SUMO) and actions (i.e., route choices) of all agents. The detailed snapshot of the traffic system is the state $s \in S$, part of which agents may observe before acting. At the end of the episode, SUMO yields individual travel times, from which the reward is calculated.

This can be formalized as:

$$\langle S, \mathcal{N}, (A_i)_{i \in \mathcal{N}}, P, (R_i)_{i \in \mathcal{N}}, (O_i)_{i \in \mathcal{N}}, v \rangle,$$

Where:

- **State space** $S$: Global state $s \in S$ encodes all relevant information about the traffic system in a given timestep, including the status of traffic lights, routes chosen by agents with earlier departure times, active vehicles, and their attributes (e.g., their trajectories, velocities, locations). States are not fully observable to the agents, so they receive only observation signals $o_i \in O_i$.
- **Agent set** $\mathcal{N}$: Human drivers and CAVs with predefined origins, destinations, and departure times.
- **Action spaces** $(A_i)_{i \in \mathcal{N}}$: Set of $K_i$ precomputed routes connecting the agent $i$'s origin and destination: $A_i = \{0, ..., K_i - 1\}$.
- **Transition function** $P$: It is a result of interplay between the agents actions (i.e. route choices) and the dynamics of the traffic flow (in URB simulated with SUMO), which updates the global state $S$ at every timestep by progressing the vehicles along their routes towards destination according to the network topology and Intelligent Driver Model (IDM) implemented in SUMO [62].
- **Reward functions** $(R_i)_{i \in \mathcal{N}}$: Computed for each agent from individual travel times at the end of the episode. For human drivers, it is the (negative) travel time. For CAVs, it can be any linear combination of own, group, and system travel time statistics. The parameterization of this combination models the CAV behavior, which can be selfish, social, altruistic, malicious, etc. (see [56] for classification of CAV behaviors).
- **Observation functions** $(O_i)_{i \in \mathcal{N}}$: It is assumed that each CAV agent, before making a routing decision, receives information about the route choices made in the current episode by agents whose departure times are earlier than the departure of the observing agent. Meanwhile, human agents make decisions solely based on their cost expectations (see Appendix E.3).
- **Agent selection mechanism** $\upsilon$: Agents act sequentially following a temporal order of departure times within an episode.

Such formulation allows us to apply a wide variety of solution methods (not only MARL algorithms). Additionally, it offers flexibility in the task formulation (e.g., in the reward, observation, or learning loop) to cover most relevant instances of URB problems. Unlike Markov Decision Process (MDP) and Partially Observable Stochastic Game (POSG) models (as used in [58, 61, 15]), AEC provides an intuitive interface for implementing a decision-making process within a turn-based structure, based on partial observations of real-time traffic conditions.

## 2.4 Multi-agent reinforcement learning algorithms

We benchmark four promising state-of-the-art MARL algorithms, which are applicable to discrete action space scenarios and are common in baselines or building blocks of more complex algorithms [21, 18, 51, 66, 73, 54]. For investigating the properties of solutions found by different classes of algorithms, we use two *independent learning* (IL) algorithms, where each agent learns separately and treats the other agents as part of the environment, and two algorithms with the *Centralized Training Decentralized Execution* (CTDE) property [4], where agents learn decentralized policies from local observations while collectively maximizing a shared global objective using global state information in centralized training. IL introduces non-stationarity from the perspective of each agent, often resulting in a lack of convergence guarantees in multi-agent settings [59], yet typically requires less computational resources and is easier to scale to large environments [38]. CTDE, on the other hand, helps address the non-stationarity that arises in the IL case [74].

- **Independent Q-Learning (IQL)** [59] is a value-based IL algorithm where each agent trains its own deep Q network. It is a useful reference point as it often works well in practice; nevertheless, it lacks theoretical convergence guarantees [19, 41].
- **Independent Proximal Policy Optimisation (IPPO)** [16] is an actor-critic method that has empirically been shown to be effective in a variety of tasks [70, 51].
- **Multi-Agent PPO (MAPPO)** [70] is an actor-critic algorithm that utilizes a centralized critic (in contrast to IPPO). It is considered a competitive baseline for cooperative MARL tasks.
- **QMIX** [55] employs a mixing network to decompose the joint state-action value function. QMIX is a *Value Decomposition* method, designed for fully cooperative tasks, where all agents share a common reward. Value decomposition methods learn a centralized joint state-action value function and factorize it into individual agent-specific value functions to enable decentralized execution while attributing each agent's contribution to the collective reward.

# 3   Urban Routing Benchmark - URB

URB is a collection of real-world traffic networks and associated realistic demand patterns, paired with reusable experiment scripts and a parameterization scheme. A URB experiment can be specified and initialized with a simple command:

```
python scripts/<script_name> --id <exp_id> --alg-conf <hyperparam_id>
    --env-conf <env_conf_id> --task-conf <task_id> --net <net_name>
    --env-seed <env_seed> --torch-seed <torch_seed>
```

Above:

- `<script_name>` points to the experiment scripts, which may be one of our baselines, provided algorithm implementations, or custom.
- `<id>` is the unique experiment identifier.
- `<hyperparam_id>`, `<env_conf_id>`, and `<task_id>` control the algorithm hyperparameterization, experiment settings (e.g., action space, disk operations), and task specifications (e.g., the share of CAVs and their reward), respectively.
- `<net_name>` is the name instance (network graph and corresponding demand pattern).
- `<env_seed>` and `<torch_seed>` control the reproducibility of the environment and training, respectively.

Training records and plots are saved in `results/<exp_id>`. We document the installation and usage of URB in Appendix C.3.

URB's workflow is integrated with the standard ML and transportation tools and libraries. Day-to-day route choices of humans and CAVs in mixed traffic are simulated using RouteRL [3], a framework that bridges MARL with a microscopic traffic simulation (SUMO). SUMO (detailed in Appendix E.2) is an open-source agent-based traffic simulation tool that reproduces how individual vehicles traverse the complex traffic networks [40]. An experiment in URB is defined through standardized input formats: CSV files, OpenStreetMap graphs [49], and JSON editable configuration files. For route generation, we use our implementation of Dial-like route generator JanuX [1] (detailed in Appendix E.4).

**Problem instance: Road network and demand pattern**   URB task is executed on a given network with a given demand pattern. URB is shipped with traffic networks of 28 Île-de-France subregions and Ingolstadt (from RESCO). Apart from the road networks, URB comes with realistic trip demand patterns associated with each network, in the format of sets of agents defined with their origins, destinations, and departure times (we use AM peak from daily demand data). We use external demand patterns (like [5] for Ingolstadt) or a synthetic demand generation pipeline based on empirical data (like [24] for 28 Île-de-France networks). The set of simulated agents is stored in the readable `agents.csv` files within the provided dataset. More details on network extraction and demand generation are documented in Appendix A. The three traffic networks, which are used in *Scenario 1* reported in Section 4.1, are visualized in Figure 2. We also provide the visualizations and demand statistics associated with each network included in URB in Appendix F.

URB **training pipeline:**   A typical task starts with a few hundred days of human learning, where each episode is treated as a single day of commute. This is followed by a *mutation* where a given share of vehicles become CAVs. Then, humans do not learn (but act according to already learned policies), and the CAVs start training. Here, episodes are virtual and do not necessarily have physical meaning. Finally, we run rollouts to showcase learned routing strategies, where both groups of agents follow their fixed policies for choosing their routes.

**Scenarios**   To thoroughly test candidate strategies, URB facilitates experimenting with a wide range of route-choice tasks. URB is modular and highly customizable, with a parameterization scheme allowing configuration of:

- **Algorithms:** Off-the-shelf and custom implementations, including TorchRL's modular components for building policies, value functions, and losses, as well as multi-agent implementations where agent policies can be centralized or decentralized, and agents may share information.

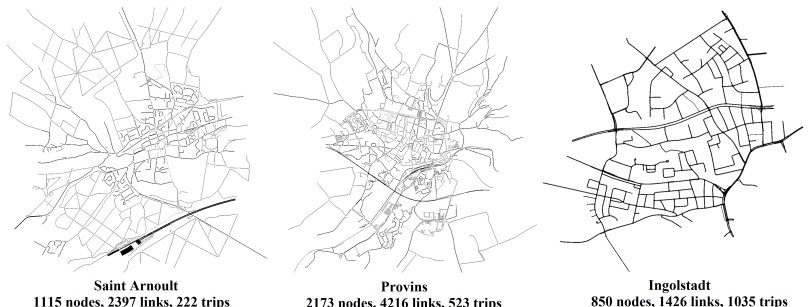

**Saint Arnoult**
1115 nodes, 2397 links, 222 trips

**Provins**
2173 nodes, 4216 links, 523 trips

**Ingolstadt**
850 nodes, 1426 links, 1035 trips

Figure 2: The traffic networks used in our benchmarking study (Section 4.1), shown in order of increasing demand levels: (a) St. Arnoult (small), (b) Provins (medium) and (c) Ingolstadt (large; from RESCO traffic light benchmark [5]).

- **Network topology**: One of 29 real-world traffic networks and associated demand data, varying in graph size and congestion, to test generalization and scalability as problem complexity increases.
- **CAV market share**: Fraction of agents that become CAVs, which can range from 0 up to 100% of agents.
- **CAV behavior profile**: Users can choose the objective to be pursued by the CAV fleet. These behaviors (as mentioned in Section 2.3) allow assessing the different social and ethical consequences of CAV deployment.
- **Human learning model**: Users can choose and parameterize the human learning model according to the driver behavior they wish to model in a given city (detailed in Appendix E.3).
- **Action spaces**: Users can set the choice set size (number of routes).
- **Length and phases of the experiment** and **hyperparameterization** for the selected algorithm.

**Evaluation metrics**  The core metric is the travel time $t$, which is both the core term of the utility for human drivers (rational utility maximizers) and of the CAVs' reward. To compare the general impact of the CAVs on the system, we calculate the average times over all agents ($t^{\text{pre}}$, $t^{\text{train}}$, $t^{\text{test}}$) by the end of human learning, CAV training, and policy testing, respectively. The $t_{\text{CAV}}$ and $t_{\text{HDV}}$ are the resulting travel times of CAVs and humans in the test phase. We measure the *cost of training*, computed as: $c_{\text{all}} = \sum_i \sum_{\tau \in \mathcal{S}_{\text{train}}} (t_i^\tau - t_i^{\text{pre}})/(|\mathcal{N}| \cdot |\mathcal{S}_{\text{train}}|)$, where $t_i^\tau$ is the travel time of agent $i$ in episode $\tau$, $t_i^{\text{pre}}$ is the average experienced time of agent $i$ for the last 50 days before CAVs are introduced, and $\mathcal{S}_{\text{train}}$ is the sequence of CAV training episodes. Similarly, we introduce $c_{\text{HDV}}$ and $c_{\text{CAV}}$ for the respective groups of agents. To better understand the causes of the changes in travel time, we track the changes in mean speed $\boldsymbol{\Delta}_{\text{V}}$ and mileage $\boldsymbol{\Delta}_{\text{L}}$ (extracted from SUMO) for the policy testing period (compared to pre-CAV). The winrate **WR** is the percentage of experiment runs where CAVs, after training, reached shorter mean travel times than humans ($t_{\text{CAV}} < t^{\text{pre}}$).

**Baselines**  URB includes naive baseline methods for the route choice problem for groundedness in the benchmarking tasks. **All-or-Nothing** (AON) model deterministically selects the route with the minimal free-flow time (expected travel time with no congestion). **Random** model selects an action randomly with a uniform probability. The **human** baseline assumes that CAVs replicate human routing decisions (apply trained human model).

## 4 Results

### 4.1 Scenario 1: Mixed autonomy

We report the result of the most representative scenario (out of multiple possible with URB) performed on three networks (see Figure 2).

**Scenario 1:** In a given network with a fixed demand pattern, experienced human agents have learned their route-choice strategies (minimized travel times). At some point, a 40% share of drivers *mutate* to CAVs and delegate their routing decisions. Then, for a given number of training episodes, the agents develop routing strategies to minimize their delay using MARL.

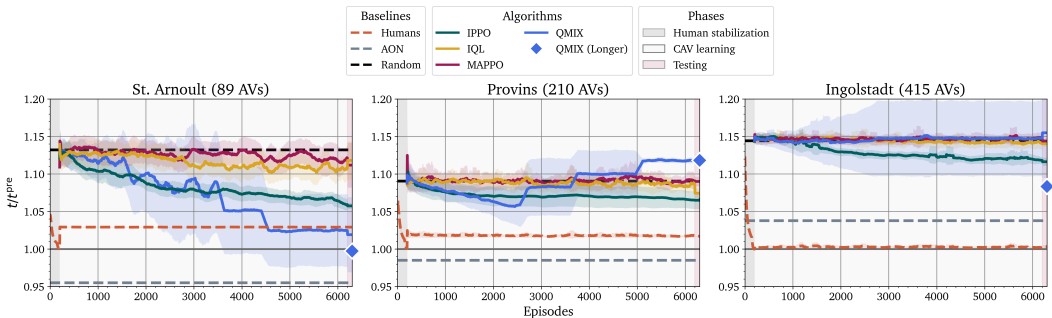

Figure 3: Mean travel times normalized by the pre-CAV mean human travel times ($t/t^{pre}$) across episodes in 3 instances for Scenario 1. Each plot visualizes the averages of five seeded repetitions, along with 95% confidence intervals. Smoothing was applied using a moving average of 150 episodes. Human travel times (orange dashed) report the mean human travel times averaged over all experiments in that instance. Background patches indicate phases: 200, 6 000, and 100 episodes (days simulated) for the human stabilization, CAV training, and policy testing, respectively. We conduct an additional training with QMIX for 20 000 training episodes (blue diamond). Many algorithms hardly beat the random baseline. Only QMIX on the smallest instance (St. Arnoult) managed to outperform humans, though not consistently, as indicated by the large variability across trials.

We use cooperative rewards (minimizing group travel time) for centralized training algorithms (MAPPO, QMIX) and selfish rewards (minimizing own travel time) for IL algorithms (IQL, IPPO). We visualize and assess (using URB metrics) our results in Figure 3 and Table 1, respectively. All experiment parameterizations and computation times are detailed in Appendix B.

Table 1: Scenario 1 results for three cities (mean ± std over five seeded runs). Pre-CAV mean travel times ($t^{pre}$) are constant per network: **St. Arnoult: 3.15, Provins: 2.8, Ingolstadt: 4.21**. For each city, the best of each metric is **bolded**, while the best of the RL algorithms is underlined. Not only do the CAVs experience a longer travel time $t_{CAV}$ than in the human-only system $t^{pre}$, but the human agents ($t_{HDV}$) are also disadvantaged by the CAV deployment. Training costs **c** are significant for all instances, and overall network performance decreased (lower mean speed: $\Delta_V < 0$ and increased mileage: $\Delta_L > 0$). Out of all algorithms, only QMIX managed to outperform humans in St. Arnoult, while in more congested systems, it performed even worse than the random baseline. IQL and MAPPO failed to converge and reached performance nearly at random.

| | | $t^{TEST}$ | $t_{CAV}$ | $t_{HDV}$ | $c_{ALL}$ | $c_{HDV}$ | $c_{CAV}$ | $\Delta_V$ | $\Delta_L$ | WR |
|---|---|---|---|---|---|---|---|---|---|---|
| **St. Arnoult** | IPPO | 3.28 (0.004) | 3.33 (0.013) | 3.25 (0.008) | 0.63 (0.015) | 0.13 (0.004) | 1.38 (0.034) | -0.24 (0.067) | 0.06 (0.004) | 0% |
| | IQL | 3.36 (0.040) | 3.53 (0.104) | 3.24 (0.005) | 0.66 (0.000) | 0.14 (0.000) | 1.44 (0.004) | -0.37 (0.115) | 0.09 (0.021) | 0% |
| | MAPPO | 3.35 (0.049) | 3.51 (0.121) | 3.25 (0.004) | 0.66 (0.000) | 0.14 (0.004) | 1.45 (0.000) | -0.27 (0.129) | 0.09 (0.019) | 0% |
| | QMIX | 3.24 (0.080) | 3.21 (0.206) | 3.25 (0.004) | 0.65 (0.004) | 0.14 (0.005) | 1.43 (0.005) | -0.22 (0.034) | 0.03 (0.040) | 80% |
| | HUMAN | **3.15** | N/A | **3.15** | N/A | N/A | N/A | **0.00** | **0.00** | **100%** |
| | AON | **3.15** | **3.01** | 3.25 | **0.55** | **0.09** | **1.21** | -0.06 | **0.00** | **100%** |
| | RANDOM | 3.38 | 3.58 | 3.25 | 0.60 | **0.09** | 1.36 | -0.33 | 0.10 | 0% |
| **Provins** | IPPO | 2.90 (0.015) | 2.98 (0.040) | 2.85 (0.004) | 0.61 (0.271) | 0.31 (0.217) | 1.05 (0.356) | -0.52 (0.080) | 0.05 (0.009) | 0% |
| | IQL | 2.91 (0.011) | 3.01 (0.027) | 2.85 (0.008) | 1.40 (0.104) | 0.92 (0.068) | 2.12 (0.183) | -0.58 (0.093) | 0.05 (0.007) | 0% |
| | MAPPO | 2.93 (0.011) | 3.05 (0.024) | 2.84 (0.005) | 1.29 (0.162) | 0.83 (0.110) | 2.00 (0.247) | -0.69 (0.038) | 0.06 (0.004) | 0% |
| | QMIX | 2.96 (0.005) | 3.14 (0.000) | 2.85 (0.000) | 0.85 (0.215) | 0.52 (0.176) | 1.35 (0.278) | -0.82 (0.033) | 0.08 (0.000) | 0% |
| | HUMAN | **2.80** | N/A | **2.80** | N/A | N/A | N/A | **0.00** | **0.00** | **100%** |
| | AON | 2.81 | **2.76** | 2.84 | **0.47** | **0.19** | 0.99 | -0.14 | **0.00** | **100%** |
| | RANDOM | 2.93 | 3.04 | 2.85 | 0.51 | 0.22 | **0.95** | -0.62 | 0.06 | 0% |
| **Ingolstadt** | IPPO | 4.41 (0.005) | 4.71 (0.030) | 4.21 (0.023) | 2.42 (0.497) | 1.90 (0.505) | 3.19 (0.495) | -0.52 (0.095) | 0.06 (0.004) | 0% |
| | IQL | 4.46 (0.009) | 4.81 (0.024) | 4.23 (0.009) | 2.54 (0.546) | 1.93 (0.533) | 3.44 (0.562) | -0.69 (0.067) | 0.07 (0.000) | 0% |
| | MAPPO | 4.45 (0.011) | 4.82 (0.019) | 4.21 (0.008) | 2.76 (0.599) | 2.16 (0.622) | 3.66 (0.562) | -0.72 (0.066) | 0.07 (0.004) | 0% |
| | QMIX | 4.50 (0.140) | 4.87 (0.325) | 4.24 (0.015) | 1.83 (0.749) | 1.27 (0.710) | 2.67 (0.810) | -0.97 (0.235) | 0.06 (0.045) | 0% |
| | HUMAN | **4.21** | N/A | **4.21** | N/A | N/A | N/A | **0.00** | 0.00 | **100%** |
| | AON | 4.29 | **4.37** | 4.23 | **0.87** | 0.55 | **0.24** | -0.45 | **-0.01** | 0% |
| | RANDOM | 4.45 | 4.81 | 4.22 | 0.99 | **0.49** | 1.74 | -0.68 | 0.07 | 0% |

The learning of IQL exhibits the well-known instability issues of IL settings [48]. IPPO shows gradual improvements, indicating the previously explored effectiveness of value-clipping and on-policy updates for non-stationarity [70] over IQL. Nonetheless, both IL algorithms fail to achieve near-human performance, with increasing gaps in saturated networks. MAPPO and QMIX utilize centralized training mechanisms. MAPPO learning is inefficient and worsens with the increasing fleet

Table 2: Sensitivity analysis with Scenario 1 in St. Arnoult. We analyze the impact of URB's realism constraints on learning performance by testing IQL and MAPPO algorithms with varying trip demands and observation types. $t^{\mathrm{pre}}$ and $t_{\mathrm{CAV}}$ are the mean travel times of humans (before CAVs) and CAVs (after 1 200 training episodes), respectively. $\Delta_\% t^{\mathrm{pre}}$ is the CAV travel time improvement rate. In both centralized and independent learning settings, learning performance degrades with increased demand and fleet size. Moreover, we show that global observations (city-wide route selection history) lead to improved learning performance.

| EXPERIMENT | $t^{\mathrm{PRE}}$ | $t_{\mathrm{CAV}}$ | $\Delta_\% t^{\mathrm{PRE}}$ |
|---|---|---|---|
| IQL (HALF DEMAND) | 3.27 | 3.45 | $-5.50\%$ |
| IQL (ORIGINAL) | 3.15 | 3.53 | $-12.06\%$ |
| IQL (DOUBLE DEMAND) | 3.24 | 5.81 | $-79.32\%$ |
| IQL (GLOBAL OBSERVATIONS) | 3.15 | 3.26 | $-3.49\%$ |
| MAPPO (HALF DEMAND) | 3.27 | 3.44 | $-5.20\%$ |
| MAPPO (ORIGINAL) | 3.15 | 3.45 | $-9.52\%$ |
| MAPPO (DOUBLE DEMAND) | 3.24 | 5.23 | $-61.42\%$ |

size. We hypothesize that this is the result of the difficulty in handling the large global information with a centralized critic [70]. For QMIX, CAVs inconsistently beat the human baseline in St. Arnoult. Interestingly, QMIX exhibits abrupt performance jumps. This aligns with the prior empirical findings with how every agent switches their expectations at once when the QMIX hyper-network's non-negative weights realign the monotonic mixing so a different joint action maximizes $Q_{\mathrm{tot}}$, and steep drops occur when the same max-operator inflates over-estimated per-agent Q values [50]. This phenomenon compromises the reliability of QMIX in our setting, as evidenced by the wide error bands. The extended training with QMIX leads to improved performance in 2 instances; however, it fails to surpass the shorter training performance in Provins. Notably, the human travel times increased after the CAV deployment.

Additionally, we test these methods in a secondary scenario, where the system fully transitions into full autonomy. These results, reported in Appendix G.1, corroborate the observed practical shortcomings of these methods in urban routing.

## 4.2 Sensitivity analysis

To study the relation between the performance and the challenges in the problem at hand (such as increased non-stationarity, local observations, and credit assignment difficulty in large agent groups), we conduct a sensitivity analysis. We run Scenario 1 on the St. Arnoult network, halving and doubling the demand (number of agents), and using global observations (each agent observes the complete history of route selections made by all previously departed vehicles in that day). We test one independent learning (IQL) and one centralized learning (MAPPO) method, using the same implementations and parameterizations as in the previously reported experiments.

For each setting, we measure the CAV travel time improvement rates ($\Delta_\% t^{\mathrm{pre}}$), which is computed as $(t^{\mathrm{pre}} - t_{\mathrm{CAV}})/t^{\mathrm{pre}} \times 100$. The results (see Table 2) show that the algorithmic performance decreases with the demand level (from -5.5% to -79.3% for IQL, and from -5.2% to -61.4% for MAPPO) and restriction of global information (-3.5% vs -12.1% for IQL). This suggests that the locality of observations, higher demand (congestion, source of non-stationarity), and the larger size of the coordinating agent group (difficulty in credit assignment) negatively impact the algorithmic performance in our setting. In URB tasks, these complexities (realism restrictions in access to global information, fleet sizes on the order of hundreds) are inevitable and yet to be addressed by methodological advancements.

## 5 Conclusions, limitations, broader impact and future work

In this paper, we introduced a framework for testing RL routing algorithms in city-scale networks and reported a comprehensive benchmarking study using community implementations of a selection of standard MARL algorithms.

Our results show that community implementations of SOTA algorithms underperform in practical settings and are highly sensitive to realism constraints. This indicates that reaching the far-reaching objectives in urban mobility will require substantial methodological progress in future work and possibly a re-evaluation of widely adopted practices (like we report in Appendix G.2, adjustments to community implementations yield notable performance gains). This reinforces the need for a community benchmark, where convincing answers to the open research questions can be developed through a series of structured experiments. With URB, we aimed to fill this gap and establish a ground for reliable testing of future advancements triggered by the methodological shortcomings uncovered by URB.

Some of the **limitations** of this work are: i) The driving model is the same for humans and CAVs; we intentionally neglect the promised traffic flow improvements of CAVs to isolate their impact on route-choice only. ii) The routes are chosen by agents once per episode. An even more challenging scenario would involve adjusting the route dynamically, making it a multi-decision setting (in contrast to one per episode). iii) The demand is fixed concerning OD pair and departure times, while in real cities, this is not the case. This would add noise to the environment and render it even less stationary. iv) We consider a single CAV fleet, while multiple competing providers are equally possible in the future. Finally, v) an experimental scheme is limited and ought to be widened (to cover more scenarios, algorithms, and instances) and deepened (to better tailor the most promising algorithms for this problem).

These limitations result from our deliberate focus on an isolated setting to attribute disturbances solely to the coexistence of humans and automated (possibly coordinated) decision-making systems in shared traffic. This controlled design removes confounds from richer setups and allows us to examine how mixed route choice shapes urban mobility dynamics. Nonetheless, our ultimate objective is to lay the groundwork for more elaborate analyses, and with URB, we establish a challenging and extensible starting point. URB is built to accommodate problem extensions, which the community will hopefully address as soon as the core issues are resolved and foundational knowledge is gathered on the simpler tasks introduced here.

Solving the urban routing problem by CAVs may have a **broader societal impact** such as: i) Potentially reduced travel times, congestion, and emissions if the developed algorithms are used properly. ii) Undesirable use by CAV operators to boost profits while exploiting human drivers. iii) Deterioration of driving conditions for human drivers in cities if the algorithms are not designed or used cautiously. iv) Pressure on human drivers to join commercial collective routing schemata as independent driving becomes inefficient.

The **future work** could encompass i) Extending the benchmark by addressing the issues raised in the limitations above. ii) Improving implementations of RL algorithms and developing alternative RL and non-RL algorithms to challenge the benchmark leaderboard. iii) Identification of the fundamental reasons why some algorithms performed poorly in certain settings to improve them and advance the RL theory. iv) Inclusion of more socially aware or task-specific metrics like equity or fuel emissions after the models are calibrated and validated on those metrics.

Predicting the future of autonomous driving is not an easy task. The safety issues are the most urgent concern, once they are solved, higher-level considerations such as collective routing may become more relevant. Recognizing this, we hope that the presented benchmark will contribute to reducing the uncertainty related to the introduction of fleets of CAVs on a large scale and preempting and mitigating any problematic scenarios this may involve.

**Acknowledgement**    This work was financed by the European Union within the Horizon Europe Framework Programme (ERC Starting Grant COeXISTENCE no. 101075838). Views and opinions expressed are however those of the authors only and do not necessarily reflect those of the European Union or the European Research Council Executive Agency. Neither the European Union nor the granting authority can be held responsible for them.

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

# Checklist

1. **Claims**

   Question: Do the main claims made in the abstract and introduction accurately reflect the paper's contributions and scope?

   Answer: [Yes]

   Justification: Yes, the main claims made in the abstract accurately reflect the contributions and scope of the paper. The abstract clearly identifies the motivation — CAVs' potential for congestion reduction through collective, RL-based routing — and acknowledges the current gap in standardized, realistic benchmarks for evaluating such approaches. It then introduces URB, the Urban Routing Benchmark, as a contribution addressing this gap. In particular, URB acts as a fair and critical assessment ground for algorithms, showing where they may perform or fall short in practical use-cases. Furthermore, it does not overstate the performance of MARL algorithms; rather, it emphasizes their limitations, setting realistic expectations for the reader. The provided metrics uncover the different dimensions of the impact of CAVs on the system and highlight areas for improvement. Overall, the claims are well-grounded and align with the paper's contributions as outlined, and we strive to fulfill these promises in the body of the manuscript.

   Guidelines:

   - The answer NA means that the abstract and introduction do not include the claims made in the paper.
   - The abstract and/or introduction should clearly state the claims made, including the contributions made in the paper and important assumptions and limitations. A No or NA answer to this question will not be perceived well by the reviewers.
   - The claims made should match theoretical and experimental results, and reflect how much the results can be expected to generalize to other settings.
   - It is fine to include aspirational goals as motivation as long as it is clear that these goals are not attained by the paper.

2. **Limitations**

   Question: Does the paper discuss the limitations of the work performed by the authors?

   Answer: [Yes]

   Justification: We report the specific problem addressed with our benchmark and discuss its limitations beyond (toward alternative formulations). We included limitations of our experimental results, limited to a few instances (3 cities), and selected algorithms in specific parameterizations. Additionally, all assumptions made by components used for the experiments are detailed in the paper and in the Appendix (see Appendix E). In the final section, we discuss potential future steps that may overcome the limitations described above.

   Guidelines:

   - The answer NA means that the paper has no limitation while the answer No means that the paper has limitations, but those are not discussed in the paper.
   - The authors are encouraged to create a separate "Limitations" section in their paper.
   - The paper should point out any strong assumptions and how robust the results are to violations of these assumptions (e.g., independence assumptions, noiseless settings, model well-specification, asymptotic approximations only holding locally). The authors should reflect on how these assumptions might be violated in practice and what the implications would be.
   - The authors should reflect on the scope of the claims made, e.g., if the approach was only tested on a few datasets or with a few runs. In general, empirical results often depend on implicit assumptions, which should be articulated.
   - The authors should reflect on the factors that influence the performance of the approach. For example, a facial recognition algorithm may perform poorly when image resolution is low or images are taken in low lighting. Or a speech-to-text system might not be used reliably to provide closed captions for online lectures because it fails to handle technical jargon.

- The authors should discuss the computational efficiency of the proposed algorithms and how they scale with dataset size.
- If applicable, the authors should discuss possible limitations of their approach to address problems of privacy and fairness.
- While the authors might fear that complete honesty about limitations might be used by reviewers as grounds for rejection, a worse outcome might be that reviewers discover limitations that aren't acknowledged in the paper. The authors should use their best judgment and recognize that individual actions in favor of transparency play an important role in developing norms that preserve the integrity of the community. Reviewers will be specifically instructed to not penalize honesty concerning limitations.

3. **Theory assumptions and proofs**

Question: For each theoretical result, does the paper provide the full set of assumptions and a complete (and correct) proof?

Answer: [NA]

Justification: This paper introduces a novel benchmark for route choice, no theorems were stated in the article.

Guidelines:

- The answer NA means that the paper does not include theoretical results.
- All the theorems, formulas, and proofs in the paper should be numbered and cross-referenced.
- All assumptions should be clearly stated or referenced in the statement of any theorems.
- The proofs can either appear in the main paper or the supplemental material, but if they appear in the supplemental material, the authors are encouraged to provide a short proof sketch to provide intuition.
- Inversely, any informal proof provided in the core of the paper should be complemented by formal proofs provided in appendix or supplemental material.
- Theorems and Lemmas that the proof relies upon should be properly referenced.

4. **Experimental result reproducibility**

Question: Does the paper fully disclose all the information needed to reproduce the main experimental results of the paper to the extent that it affects the main claims and/or conclusions of the paper (regardless of whether the code and data are provided or not)?

Answer: [Yes]

Justification: Our framework with detailed usage instructions, as well as data used in all experiments, is available online in an unchanged form. Reported experiments are accompanied by reproducible code and the scripts that allow us to recreate the figures from raw results. The very purpose of our work is to achieve full reproducibility to encourage further research in route choice analysis. See Appendix B for details.

Guidelines:

- The answer NA means that the paper does not include experiments.
- If the paper includes experiments, a No answer to this question will not be perceived well by the reviewers: Making the paper reproducible is important, regardless of whether the code and data are provided or not.
- If the contribution is a dataset and/or model, the authors should describe the steps taken to make their results reproducible or verifiable.
- Depending on the contribution, reproducibility can be accomplished in various ways. For example, if the contribution is a novel architecture, describing the architecture fully might suffice, or if the contribution is a specific model and empirical evaluation, it may be necessary to either make it possible for others to replicate the model with the same dataset, or provide access to the model. In general. releasing code and data is often one good way to accomplish this, but reproducibility can also be provided via detailed instructions for how to replicate the results, access to a hosted model (e.g., in the case of a large language model), releasing of a model checkpoint, or other means that are appropriate to the research performed.

- While NeurIPS does not require releasing code, the conference does require all submissions to provide some reasonable avenue for reproducibility, which may depend on the nature of the contribution. For example
  (a) If the contribution is primarily a new algorithm, the paper should make it clear how to reproduce that algorithm.
  (b) If the contribution is primarily a new model architecture, the paper should describe the architecture clearly and fully.
  (c) If the contribution is a new model (e.g., a large language model), then there should either be a way to access this model for reproducing the results or a way to reproduce the model (e.g., with an open-source dataset or instructions for how to construct the dataset).
  (d) We recognize that reproducibility may be tricky in some cases, in which case authors are welcome to describe the particular way they provide for reproducibility. In the case of closed-source models, it may be that access to the model is limited in some way (e.g., to registered users), but it should be possible for other researchers to have some path to reproducing or verifying the results.

5. **Open access to data and code**

   Question: Does the paper provide open access to the data and code, with sufficient instructions to faithfully reproduce the main experimental results, as described in supplemental material?

   Answer: [Yes]

   Justification: The full project code is available in our GitHub repository, and the dataset (networks with trips) was uploaded to a Kaggle repository. The raw trips data used to generate our Île-de-France data is also made publicly available on a separate dataset. For the reader's reference, all the experiment data generated is versioned and shared in an open-access Zenodo data repository.

   Guidelines:
   - The answer NA means that paper does not include experiments requiring code.
   - Please see the NeurIPS code and data submission guidelines (`https://nips.cc/public/guides/CodeSubmissionPolicy`) for more details.
   - While we encourage the release of code and data, we understand that this might not be possible, so "No" is an acceptable answer. Papers cannot be rejected simply for not including code, unless this is central to the contribution (e.g., for a new open-source benchmark).
   - The instructions should contain the exact command and environment needed to run to reproduce the results. See the NeurIPS code and data submission guidelines (`https://nips.cc/public/guides/CodeSubmissionPolicy`) for more details.
   - The authors should provide instructions on data access and preparation, including how to access the raw data, preprocessed data, intermediate data, and generated data, etc.
   - The authors should provide scripts to reproduce all experimental results for the new proposed method and baselines. If only a subset of experiments are reproducible, they should state which ones are omitted from the script and why.
   - At submission time, to preserve anonymity, the authors should release anonymized versions (if applicable).
   - Providing as much information as possible in supplemental material (appended to the paper) is recommended, but including URLs to data and code is permitted.

6. **Experimental setting/details**

   Question: Does the paper specify all the training and test details (e.g., data splits, hyperparameters, how they were chosen, type of optimizer, etc.) necessary to understand the results?

   Answer: [Yes]

   Justification: We disclose the experiment configurations and all the experiment data produced in the process of tuning the hyperparameters in the dedicated data repository. We document the details on the creation of our dataset in Appendix A. The complete scripts of algorithms used are stored with the detailed code and parameterization in the public repository.

Guidelines:

- The answer NA means that the paper does not include experiments.
- The experimental setting should be presented in the core of the paper to a level of detail that is necessary to appreciate the results and make sense of them.
- The full details can be provided either with the code, in appendix, or as supplemental material.

7. **Experiment statistical significance**

Question: Does the paper report error bars suitably and correctly defined or other appropriate information about the statistical significance of the experiments?

Answer: [Yes]

Justification: For each experiment configuration, we ran five seeded and fully reproducible trials. We report the ranking of methods under different settings by comparing the metric averages of each trial and provide statistics regarding performance consistency. We do so using standard deviations in our tables and confidence intervals in our plots. These statistics provide additional context regarding the reliability of a method.

Guidelines:

- The answer NA means that the paper does not include experiments.
- The authors should answer "Yes" if the results are accompanied by error bars, confidence intervals, or statistical significance tests, at least for the experiments that support the main claims of the paper.
- The factors of variability that the error bars are capturing should be clearly stated (for example, train/test split, initialization, random drawing of some parameter, or overall run with given experimental conditions).
- The method for calculating the error bars should be explained (closed form formula, call to a library function, bootstrap, etc.)
- The assumptions made should be given (e.g., Normally distributed errors).
- It should be clear whether the error bar is the standard deviation or the standard error of the mean.
- It is OK to report 1-sigma error bars, but one should state it. The authors should preferably report a 2-sigma error bar than state that they have a 96% CI, if the hypothesis of Normality of errors is not verified.
- For asymmetric distributions, the authors should be careful not to show in tables or figures symmetric error bars that would yield results that are out of range (e.g. negative error rates).
- If error bars are reported in tables or plots, The authors should explain in the text how they were calculated and reference the corresponding figures or tables in the text.

8. **Experiments compute resources**

Question: For each experiment, does the paper provide sufficient information on the computer resources (type of compute workers, memory, time of execution) needed to reproduce the experiments?

Answer: [Yes]

Justification: Detailed information on computing power used in the experiments is provided in Appendix B. We conducted the experiments using our institution's computing infrastructure, presented in Table 4. Computation times are presented in Table 5.

Guidelines:

- The answer NA means that the paper does not include experiments.
- The paper should indicate the type of compute workers CPU or GPU, internal cluster, or cloud provider, including relevant memory and storage.
- The paper should provide the amount of compute required for each of the individual experimental runs as well as estimate the total compute.
- The paper should disclose whether the full research project required more compute than the experiments reported in the paper (e.g., preliminary or failed experiments that didn't make it into the paper).

9. **Code of ethics**

   Question: Does the research conducted in the paper conform, in every respect, with the NeurIPS Code of Ethics https://neurips.cc/public/EthicsGuidelines?

   Answer: [Yes]

   Justification: Our research conforms, in every respect, with the NeurIPS Code of Ethics.

   Guidelines:

   - The answer NA means that the authors have not reviewed the NeurIPS Code of Ethics.
   - If the authors answer No, they should explain the special circumstances that require a deviation from the Code of Ethics.
   - The authors should make sure to preserve anonymity (e.g., if there is a special consideration due to laws or regulations in their jurisdiction).

10. **Broader impacts**

    Question: Does the paper discuss both potential positive societal impacts and negative societal impacts of the work performed?

    Answer: [Yes]

    Justification: URB is a benchmark for a problem of high social impact and significance. The CAV routing problem may have a profound impact on urban traffic systems worldwide and on individual drivers in both CAVs and traditional vehicles. This impact may go beyond car users and may also negatively affect public transit users. We discuss it in the introduction and highlight its significance, while the last section discusses both the likely positive societal impact and the potential threats if the algorithms developed using the benchmark are misused. Hopefully, URB leaderboard will eventually be dominated by efficient, scalable, socially aware solutions that, when implemented on real networks, will improve the performance of all parties involved. URB will be gradually extended with problems that will arise in the future, when CAVs will start operating in our cities in various configurations, with candidate solutions (from within the RL community and outside) evaluated on a wide set of measures and problem instances. This is particularly timely before CAV fleets are deployed, to inform society about potential threats and motivate vehicle developers to propose sustainable fleet operating algorithms before they enter our cities.

    Guidelines:

    - The answer NA means that there is no societal impact of the work performed.
    - If the authors answer NA or No, they should explain why their work has no societal impact or why the paper does not address societal impact.
    - Examples of negative societal impacts include potential malicious or unintended uses (e.g., disinformation, generating fake profiles, surveillance), fairness considerations (e.g., deployment of technologies that could make decisions that unfairly impact specific groups), privacy considerations, and security considerations.
    - The conference expects that many papers will be foundational research and not tied to particular applications, let alone deployments. However, if there is a direct path to any negative applications, the authors should point it out. For example, it is legitimate to point out that an improvement in the quality of generative models could be used to generate deepfakes for disinformation. On the other hand, it is not needed to point out that a generic algorithm for optimizing neural networks could enable people to train models that generate Deepfakes faster.
    - The authors should consider possible harms that could arise when the technology is being used as intended and functioning correctly, harms that could arise when the technology is being used as intended but gives incorrect results, and harms following from (intentional or unintentional) misuse of the technology.
    - If there are negative societal impacts, the authors could also discuss possible mitigation strategies (e.g., gated release of models, providing defenses in addition to attacks, mechanisms for monitoring misuse, mechanisms to monitor how a system learns from feedback over time, improving the efficiency and accessibility of ML).

11. **Safeguards**

Question: Does the paper describe safeguards that have been put in place for responsible release of data or models that have a high risk for misuse (e.g., pretrained language models, image generators, or scraped datasets)?

Answer: [NA]

Justification: Our benchmark does not have malicious use, nor can the data be used in a way that harms people.

Guidelines:

- The answer NA means that the paper poses no such risks.
- Released models that have a high risk for misuse or dual-use should be released with necessary safeguards to allow for controlled use of the model, for example by requiring that users adhere to usage guidelines or restrictions to access the model or implementing safety filters.
- Datasets that have been scraped from the Internet could pose safety risks. The authors should describe how they avoided releasing unsafe images.
- We recognize that providing effective safeguards is challenging, and many papers do not require this, but we encourage authors to take this into account and make a best faith effort.

12. **Licenses for existing assets**

Question: Are the creators or original owners of assets (e.g., code, data, models), used in the paper, properly credited and are the license and terms of use explicitly mentioned and properly respected?

Answer: [Yes]

Justification: The paper, codebase, and the dataset have corresponding licenses added to their repositories. Licensing and availability details are provided in Appendix C.1.

Guidelines:

- The answer NA means that the paper does not use existing assets.
- The authors should cite the original paper that produced the code package or dataset.
- The authors should state which version of the asset is used and, if possible, include a URL.
- The name of the license (e.g., CC-BY 4.0) should be included for each asset.
- For scraped data from a particular source (e.g., website), the copyright and terms of service of that source should be provided.
- If assets are released, the license, copyright information, and terms of use in the package should be provided. For popular datasets, `paperswithcode.com/datasets` has curated licenses for some datasets. Their licensing guide can help determine the license of a dataset.
- For existing datasets that are re-packaged, both the original license and the license of the derived asset (if it has changed) should be provided.
- If this information is not available online, the authors are encouraged to reach out to the asset's creators.

13. **New assets**

Question: Are new assets introduced in the paper well documented and is the documentation provided alongside the assets?

Answer: [Yes]

Justification: Our codebase and data are well documented. We made every effort to provide a tool that is easy to use for future research. The README of the repository contains a step-by-step guide to using `URB`, which executes the same scripts we used in the paper.

Guidelines:

- The answer NA means that the paper does not release new assets.
- Researchers should communicate the details of the dataset/code/model as part of their submissions via structured templates. This includes details about training, license, limitations, etc.

- The paper should discuss whether and how consent was obtained from people whose asset is used.
- At submission time, remember to anonymize your assets (if applicable). You can either create an anonymized URL or include an anonymized zip file.

14. **Crowdsourcing and research with human subjects**

Question: For crowdsourcing experiments and research with human subjects, does the paper include the full text of instructions given to participants and screenshots, if applicable, as well as details about compensation (if any)?

Answer: [NA]

Justification: This article does not involve any research with crowdsourcing or human subjects.

Guidelines:

- The answer NA means that the paper does not involve crowdsourcing nor research with human subjects.
- Including this information in the supplemental material is fine, but if the main contribution of the paper involves human subjects, then as much detail as possible should be included in the main paper.
- According to the NeurIPS Code of Ethics, workers involved in data collection, curation, or other labor should be paid at least the minimum wage in the country of the data collector.

15. **Institutional review board (IRB) approvals or equivalent for research with human subjects**

Question: Does the paper describe potential risks incurred by study participants, whether such risks were disclosed to the subjects, and whether Institutional Review Board (IRB) approvals (or an equivalent approval/review based on the requirements of your country or institution) were obtained?

Answer: [NA]

Justification: This article does not involve any research with human subjects.

Guidelines:

- The answer NA means that the paper does not involve crowdsourcing nor research with human subjects.
- Depending on the country in which research is conducted, IRB approval (or equivalent) may be required for any human subjects research. If you obtained IRB approval, you should clearly state this in the paper.
- We recognize that the procedures for this may vary significantly between institutions and locations, and we expect authors to adhere to the NeurIPS Code of Ethics and the guidelines for their institution.
- For initial submissions, do not include any information that would break anonymity (if applicable), such as the institution conducting the review.

16. **Declaration of LLM usage**

Question: Does the paper describe the usage of LLMs if it is an important, original, or non-standard component of the core methods in this research? Note that if the LLM is used only for writing, editing, or formatting purposes and does not impact the core methodology, scientific rigorousness, or originality of the research, declaration is not required.

Answer: [NA]

Justification: LLMs were used only for writing, editing, or formatting purposes and does not impact the core methodology, scientific rigor, or originality of the research.

Guidelines:

- The answer NA means that the core method development in this research does not involve LLMs as any important, original, or non-standard components.
- Please refer to our LLM policy (https://neurips.cc/Conferences/2025/LLM) for what should or should not be described.

# Appendix

## A   Data generation

### A.1   Synthetic population with travel demand in Île-de-France networks

To ensure our agents operate within a realistic setting, we generated a synthetic population and its travel demand following the methodology of Hörl and Balac[24]. Their framework leverages publicly accessible datasets and open-source code [4], ensuring that simulation inputs can be fully reproduced. It utilizes very detailed French statistical data, which consists of granular information on individual attributes and travel behaviors.

The synthetic-population generation proceeded through four main stages:

1. **Population Synthesis.**
   We began by matching 30% microsample census data (covering disaggregated data on persons and households [28]) with population counts and characteristics for each spatial unit in Île-de-France ([27], [25], [31]). Households were replicated until the region's 12 million inhabitants were properly represented in each area. In the end, each synthetic individual had demographic attributes (age, sex, socio-professional category, employment or education status) as well as household characteristics (household size, number of vehicles, and home location in the specific area).

2. **Activity-Trip Enrichment.**
   Next, we assigned each person a full-day activity schedule and trip chain drawn from the national household travel survey ([45]). Using a statistical-matching procedure, we paired synthetic individuals with the survey's disaggregated records based on discrete attribute

---

[4]`https://github.com/eqasim-org/ile-de-france/tree/v1.2.0`

similarity (e.g., age group, sex), thereby transferring realistic activity–trip patterns to each agent.

3. **Primary Location Allocation.**
   For each household, a random home address location was sampled within its spatial zone ([26]). We then assigned workplaces to employed agents to reconcile the census-derived inter-zonal commuting matrix ([29]), enterprise locations weighted by number of employees ([14]), and the individual's target commute distance (inferred from their trip chain). Educational institutions were similarly allocated for students ([30]), drawing upon the national Service and Facility database ([32]).

4. **Secondary Activity Assignment.**
   Finally, secondary purposes (leisure, shopping, and other activities) were allocated using the approach of Hörl and Axhausen ([23]), which accounts for each agent's primary activities, proximity to service and facility locations, and matching inter-site distances along with the activity chain distances.

This procedure yielded synthetic trip data for the Île-de-France region, within which we selected inner trips in 28 subregions. This data is formatted as CSV files, each row describing a single trip made by an individual and consisting of 21 attributes of each trip, including: `person_id`, `trip_index`, `departure_time`, `arrival_time`, `preceding_purpose`, `following_purpose`, `ox`, `oy`, `dx`, `dy`, `abm_region`, ... where (`ox`, `oy`) and (`dx`, `dy`) are coordinates of the trip origin and destination.

We then converted this raw trip data into a format compatible with RouteRL using the processing pipeline described below.

1. **Demand filtering** For each region, trips are filtered according to their departure times by a predefined time window. For managing the computation time of our experiments, we set it to be a half-hour time period starting from 9 AM.

2. **Network extraction** For each region, the corresponding OpenStreetMap file extract is converted to SUMO network files (`*.nod.xml`, `*.edg.xml`, `*.con.xml`, `*.net.xml`, `*.rou.xml`, `*.tll.xml`, `*.typ.xml`) using `netconvert`. Non-passenger edges are eliminated.

3. **Edge mapping** Origin and destination coordinates are snapped to the nearest traversable edge midpoint, computed from node coordinates. Trips whose origin or destination corresponds to an isolated or dead-end edge are discarded.

4. **Route generation** For a trip to be used in route choice, we must be able to generate multiple routes between its origin and destination. Not all trips satisfied this condition. Therefore, we then filter out trips whose origin and destination cannot be connected to up to 4 routes by JanuX [1], which is the custom route generation tool, also used in RouteRL.

5. **Output files**
   - **Agent metadata** (`agents.csv:`): List of agents where each have ID, origin, destination, and departure time used in the simulations.
   - **Network files:** XML files ready to be loaded by SUMO.

With URB, we make the raw trip data[5] and the converted URB-usable network-demand dataset[6] publicly available for general use.

## A.2  Processing of InTAS Ingolstadt data from RESCO

We used one of the well-established SUMO scenarios, already utilized in RESCO traffic-light benchmark[5], namely "InTAS" [39]. It describes traffic within a real-world city, Ingolstadt (Germany), including road network layout and calibrated demands.

The demand in RESCO was static; thus we needed to convert it. We used the same trip demand as in the original dataset, yet converted each trip (vehicle assigned to a route at a given departure) to the request (origin, destination, and departure time). Then, we sampled four routes for each unique OD pair with JanuX [1] and filtered out OD pairs with fewer possible routes. Finally, we selected trips within a half-hour period for computational efficiency. The resulting data (demand and network files)

---

[5]`https://doi.org/10.34740/kaggle/ds/7302756`
[6]`https://doi.org/10.34740/kaggle/ds/7406751`

are provided within the same dataset as the Île-de-France data (see Appendix A.1). The script[7] used for this processing pipeline is publicly available in a public repository.

# B  Experiment details and reproducibility

**Code.**  All experimental results reported in this paper are created using the scripts and the configuration files included in URB's public GitHub repository[8].

**Results data.**  All results produced in the experiments reported in this paper, along with their parameterizations and visualization scripts, are stored on a Zenodo data repository[2] with open access [9]. This repository contains subdirectories that classify different experimental settings and is organized as listed in Table 3.

Table 3: Directory organization of publicly available experiment data.

| Directory name | Experiment | Reported in |
|---|---|---|
| scenario1 | 40% CAVs | Section 4.1 |
| scenario1_long | 40% CAVs (long training) | Section 4.1 |
| sc1_custom_imp | 40% CAVs (custom implementations) | Appendix G.2 |
| scenario2 | 100% CAVs | Appendix G.1 |
| sensitivity_analysis | Sensitivity analysis | Section 4.2 |
| demonstrative | Demonstrative | Appendix G.3 |
| hyperparam_search | Hyperparameter tuning | - |
| plotting_scripts | Scripts used for visualizations | Sections 4.1 and G.1 |

**Parameterization.**  Each experiment's data is organized within a dedicated directory in the above-mentioned dedicated data repository, semantically named after the used network, algorithm, and seed value. Each of these folders includes a `exp_config.json` file, which stores all the parameterizations used in that particular experiment. We refer the interested reader to these configuration files to ensure the reproducibility of our experiments.

**Hardware and compute time.**  Our experiments are conducted on our institution's computing nodes with resources allocated as listed in Table 4. Experiment compute time is highly dependent on the simulated scenario and parameterization. We share the computation time of 4 representative cases in Table 5.

Table 4: Summary of computational environment used for experiments.

| Component | Specification |
|---|---|
| CPU | Intel(R) Xeon(R) Gold 5122 CPU, 3.60GHz |
| GPU | NVIDIA GeForce RTX 2080 |
| RAM | 64 GB allocated per job |
| Operating system | Ubuntu 24.04.1 LTS |
| SUMO version | 1.18.0 |

# C  Accessibility and usage

## C.1  Licensing and availability

Our code and the input data are released under the MIT License. The code, the datasets, and the documentation on how to use URB are publicly available in the GitHub repository. The datasets

---

[7]`https://github.com/COeXISTENCE-PROJECT/extract_resco_demand`
[8]`https://github.com/COeXISTENCE-PROJECT/URB`
[9]`https://doi.org/10.5281/zenodo.17317056`

Table 5: Computation time of four representative experiments. The first three experiments are run for 200 human learning and 6 000 CAV training episodes (test phase omitted) with 40% CAVs. The last experiment is run for 200 human learning and 300 baseline running episodes for 40% CAVs.

| Traffic network | Algorithm | Runtime (hours) |
|---|---|---|
| St. Arnoult | QMIX | ~15.5 |
| Provins | QMIX | ~44 |
| Ingolstadt | QMIX | ~152.5 |
| Ingolstadt | AON (Baseline) | ~4.25 |

(networks and travel demand) are also released under the MIT License. All data for creating the benchmarking environment and task scenarios are based on publicly available open data. SUMO traffic simulator is licensed under the EPL-2.0 with GPL v2 or later as a secondary license option (refer to SUMO website[10] for more details).

## C.2 Quickstart: Code Ocean capsule

For a quick start on interaction with URB, we provide an executable code capsule at Code Ocean[11] that executes a concise demonstrative experiment using the QMIX algorithm in the St. Arnoult network. This environment comes with all dependencies (including SUMO) preinstalled, allowing the experiment to be reproduced with a single click via the *Reproducible Run* feature. We invite interested readers to explore this capsule to examine the experimental workflow and output formats in a fully isolated and controlled setting.

## C.3 Installation and usage

Ensure that SUMO is installed and available on the system. This procedure should be carried out separately[12].

Clone the URB repository from GitHub:

```
git clone https://github.com/COeXISTENCE-PROJECT/URB.git
```

Then, install the dependencies:

```
cd URB
pip install -r requirements.txt
```

To use URB with a reinforcement learning algorithm, run the following command:

```
python scripts/<script_name> --id <exp_id> --alg-conf <hyperparam_id>
    --env-conf <env_conf_id> --task-conf <task_id> --net <net_name>
    --env-seed <env_seed> --torch-seed <torch_seed>
```

Where:

- `<script_name>` points to the algorithm implementation (provided scripts, or the user's implementation). Provided scripts include: `ippo_torchrl`, `iql_torchrl`, `mappo_torchrl`, `qmix_torchrl` (scripts used in Sections 4.1 and G.1), `vdn_torchrl.py` (used in Appendix G.3), `ippo.py`, and `iql.py` (scripts used in Appendix G.2).
- `<id>` is the unique experiment identifier, which can be any string and is used to organize the training records and metrics alongside other experiments (e.g., `vdn_malicious_ingolstadt`).
- `<hyperparam_id>` is the algorithm hyperparameter configuration. It must correspond to a JSON filename (without extension) in `config/algo_config` directory. Provided scripts automatically select the algorithm-specific subfolder in this directory. Users can add their custom parameterizations by following the structure of the provided ones and use them similarly.

---

[10] https://eclipse.dev/sumo/
[11] https://codeocean.com/capsule/1896262/tree
[12] Instructions available at: https://sumo.dlr.de/docs/Installing/index.html

- `<env_conf_id>` is the environment configuration identifier. It must correspond to a JSON filename (without extension) in `config/env_config` directory. It is used to parameterize environment-specific processes, such as path generation, disk operations, etc. It is **optional** and by default is set to `config1`. Users can add their custom environment settings by following the structure of the provided ones and use them similarly.

- `<task_id>` is the task configuration identifier. It must correspond to a JSON filename (without extension) in `config/task_config` directory. It is used to parameterize the simulated scenario, such as the portion of CAVs, duration of human learning, CAV behavior, etc. Users can define custom tasks by following the structure of the provided definitions and use them similarly.

- `<net_name>` is the network graph and corresponding demand pattern. It must correspond to one of the subdirectory names in `networks/`. We provide all the networks used in this paper (`gretz_armainvilliers`, `ingolstadt_custom`, `nangis`, `nemours`, `provins`, and `saint_arnoult`) in this directory. Users can download the network of their choice from our dataset on Kaggle and place it in this directory, then use it similarly.

- `<env_seed>` is the seed for the traffic environment (default: 42).

- `<torch_seed>` is the seed for PyTorch (default: 42).

**Example:**

```
python scripts/qmix_torchrl.py --id sai_qmix_0 --alg-conf config3 --
    task-conf config4 --net saint_arnoult --env-seed 42 --torch-seed
    0
```

Results and plots will be saved in `results/<exp_id>`.

To run baseline models, use the following command (notice the additional `-model` flag instead of `-torch-seed`):

```
python scripts/baselines.py --id <exp_id> --alg-conf <hyperparam_id>
    --env-conf <configuration_id> --task-conf <task_id> --net <
    net_name> --env-seed <env_seed> --model <model_name>
```

Where `<model_name>` is one of: `aon`, `random` (baseline models included in URB, under `baseline_models/`), or `gawron` (a human learning model from RouteRL).

**Example:**

```
python scripts/baselines.py --id ing_aon --alg-conf config1 --task-
    conf config2 --net ingolstadt_custom --model aon
```

## C.4 Access to networks

We include only six traffic networks and associated demand data (Gretz-Armainvilliers, Ingolstadt, Nangis, Nemours, Provins, Saint-Arnoult) in URB's GitHub repository for code mobility. Users who wish to utilize the entire URB network and demand library can (1) download corresponding network folders from URB's Kaggle data repository, (2) place the network folder in the `networks/` directory, (3) use via the `-net` flag as described above.

## D   Novelty

To serve as a community tool, URB is designed to be accessible and comprehensive to cover many of the use cases to support the research in CAV routing tasks. It provides all the tools and functionalities typically included in a benchmarking framework and allows users to customize many aspects of the scenario or experiment. Additionally, URB is a novel testbed, as the problem of collective CAV routing in mixed systems involves many complexities that constitute a novel difficulty for MARL algorithms. In Table 6, we make this novelty more concrete by comparing different aspects of URB with other benchmarks established in the scientific literature.

Table 6: Comparison of URB with prominent MARL benchmarks. Only URB uses diverse, realistic traffic networks as the primary environment. BenchMARL [8] mainly includes grid-world or multi-robotic simulations and, like EPyMARL [51], focuses largely on intersection-style evaluations. RESCO [5] benchmarks traffic signal control using both realistic (city segments) and toy (e.g., 4×4 grid) networks. FLOW [65] has a single realistic network with a simple architecture.

| Benchmark | Diversity | Coverage | Extendability |
|---|---|---|---|
| **URB** 
 *Analysis of CAV impact on large-scale urban routing optimization* | • Scenarios 
 • Networks 
 • Tasks 
 • CAV behaviors 
 • Up to 6,924 agents | • 7 algorithm implementations 
 • 29 networks | • Extendable networks/scenarios 
 • Variable demand levels 
 • Custom algorithms |
| **BenchMARL** 
 *Standardized benchmarking across algorithms, models, and environments* | • Task variety 
 • Reward types 
 • Variable agent count | • 9 algorithm implementations 
 • 5 environments 
 • 2–49 tasks | • Extendable algorithm/model/task catalog |
| **EPyMARL** 
 *MARL approaches in environments of varying difficulty* | • Observability settings 
 • Difficulty variety 
 • 2–10 agents | • 9 algorithm implementations 
 • 5 environments | • New environments supported 
 • Plotting and tracking tools |
| **RESCO** 
 *Learning traffic signal control policies* | • Traffic signal agents in real-world city subnetworks 
 • Varying size networks | • 8 algorithm implementations 
 • 8 networks | • Support for new SUMO scenarios |
| **FLOW** 
 *Learning control laws for autonomous vehicles* | • 14–22 agents | • 6 networks | • Custom templates, controllers, and environments |

# E  Components and dependencies

## E.1  RouteRL

To study the routing behavior of CAVs in complex urban environments, we rely on RouteRL [3], an open-source framework that couples MARL with microscopic traffic simulation. RouteRL is designed to model daily route choices of heterogeneous driver agents, including both human drivers, emulated using behaviorally grounded models, and CAVs, modeled as MARL agents optimizing routing strategies based on predefined objectives such as travel time or system efficiency. The framework supports flexible experimentation through configurable traffic networks, CAV market shares, routing algorithms, and behavioral heterogeneity.

## E.2  SUMO

We use the Simulation of Urban MObility (SUMO) [36, 7] as the underlying microscopic traffic simulator to model realistic traffic dynamics in urban environments. SUMO is an open-source, highly portable, and extensible platform designed to simulate the movement of individual vehicles based on time-continuous, space-continuous traffic flow models. Its core simulation engine models

vehicle behavior using the Krauss car-following model [37], which includes stochastic acceleration, deceleration, and gap-keeping behavior to capture real-world driving variability. Lane-changing is handled through a rule-based model that accounts for safety, convenience, and strategic routing decisions. SUMO supports loading real-world road networks (e.g., from OpenStreetMap), making it ideal for simulations in large-scale, realistic urban scenarios. Each vehicle can be treated as an agent, with states defined by observable traffic variables (e.g., position, speed, headway) and actions corresponding to routing choices or lane changes. The simulator provides a high-frequency, low-latency API (TraCI) that enables real-time communication between the RL agent and the simulation environment. This allows agents to receive observations, perform actions, and obtain reward signals in a closed loop.

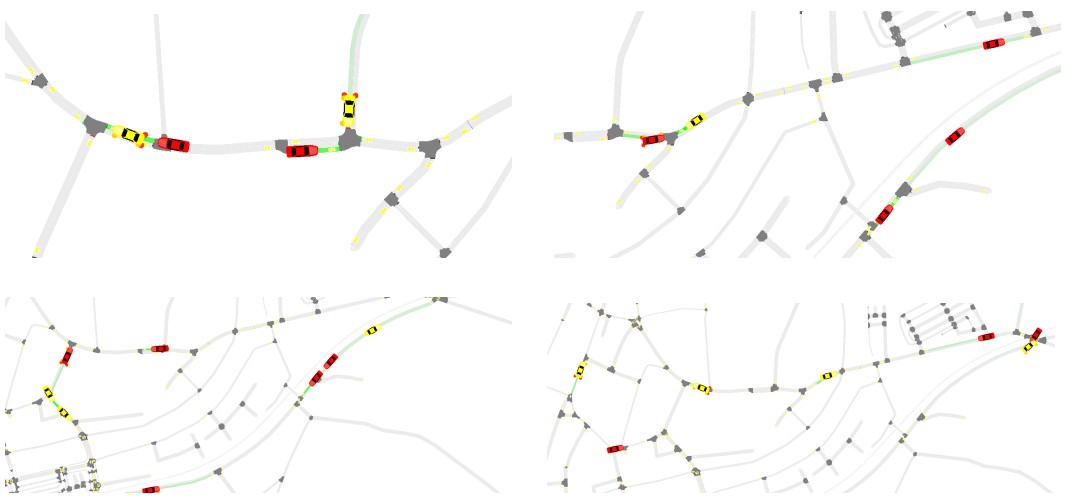

Figure 4: Screenshots from SUMO GUI, from an experiment conducted using the Provins traffic network. Yellow vehicles represent CAVs, while red vehicles indicate human drivers. Junctions are shown with dark gray, and yellow rectangles represent traffic detectors.

### E.3  Human learning model - day-to-day agent-based route choice model

**Learning**  Human agent $i$ updates expected travel time $\overline{C}_{i,\tau,k}$ every day $\tau$ on the selected route $k$, based on the actual travel time $\hat{C}_{i,j,k}$ experienced in the earlier days $j \in \{0, \ldots, \tau - 1\}$. The update, with a learning rate $\alpha_0$, occurs only if the difference between the expected and experienced travel times exceeds the bounded rationality threshold $\gamma_c$. Each day $j$ of recorded history $\tau$ is weighted with $\alpha_j$:

$$\overline{C}_{i,\tau,k} = \begin{cases} \overline{C}_{i,\tau-1,k} & \text{if } a(i, \tau - 1) \neq k \\ \begin{cases} \overline{C}_{i,\tau-1,k} \text{ if } | \overline{C}_{i,\tau-1,k} - \hat{C}_{i,\tau,k} | <= \gamma_C \\ \alpha_0 * \overline{C}_{i,\tau-1,k} + \sum_{j=0}^{\tau-1} \alpha_j * \hat{C}_{i,j,k} \end{cases} & \text{if } a(i, \tau - 1) = k \end{cases} \tag{1}$$

**Agents' decision process - act**  Based on the expected costs $\overline{C}$ learned so far, each agent selects a subjectively optimal route $a(i, \tau)$ following a utility maximization model. We model this behavior by adding a random variable $\varepsilon$ to the cost, which is multiplied by $\beta$ (to control the bias of the decisions):

$$U_{i,\tau,k} = \beta \overline{C}_{i,\tau,k} + \varepsilon \tag{2}$$

where $\varepsilon = w_i * \varepsilon_i + w_{i,k} * \varepsilon_{i,k} + W_{i,k,\tau} * \varepsilon_{i,k,\tau}$ and

$$a(i, \tau) = \begin{cases} a(i, \tau - 1) & \text{if } | \overline{C}_{i,\tau-1,k} - \overline{C}_{i,\tau,k} | <= \gamma_u \\ \begin{cases} \arg\max_{k \in K_{OD}} U_{i,\tau,k} & \text{with prob. } 1 - \delta \\ \text{random choice} & \text{with prob. } \delta \end{cases} & \text{otherwise} \end{cases} \tag{3}$$

**Remark:** While RouteRL allows complex human models covering many empirically observed behaviors (such as heterogeneous agents), the human model parameterization used in this study is a deliberate simplification of the real-world heterogeneity in human decision-making. By removing unnecessary variance, we analyze the group-level interactions under controlled conditions. Although a less sparse configuration would be useful to assess practical performance, we believe that the matter of scalability will follow the identification of solutions to overcome the current algorithmic challenges (identified in Section 4). Therefore, for our experiments, we opted for a simple model with $\gamma_C = 0$, $\gamma_u = 0$, $\varepsilon = 0$, and $\delta = 0$, which boils down to:

$$\overline{C}_{i,\tau,k} = \begin{cases} \overline{C}_{i,\tau-1,k} & \text{if } a(i,\tau-1) \neq k \\ 0.8 * \overline{C}_{i,\tau-1,k} + 0.2 * \hat{C}_{i,\tau-1,k} & \text{if } a(i,\tau-1) = k \end{cases} \tag{4}$$

and

$$a(i,\tau) = \arg\min_k \overline{C}_{i,\tau,k}.$$

The initial conditions are specified by $\overline{C}_{i,0,k}$ being the free flow travel time via route $k$, which is computed from the network graph description generated by Open Street Map.

### E.4   Route generation

For each agent, the action space is the discrete route options that connect their origins to their destinations and are precomputed. Agents with the same origin and destinations have the same action space. The action space sizes (i.e., number of routes) are the same for all agents and are determined by the parameter `number_of_paths`, which is set to 4 in all experiments reported in this study. All parameters used for path generation are stored in the `exp_config.json` in the dedicated directory for each experiment in our public experiment data repository (See Appendix B for details).

Path generation procedure is managed by RouteRL and carried out by JanuX [1], a NetworkX-compatible path generation tool. For a given traffic network and user parameters, JanuX runs a modified Dial-like [17] algorithm-based process to sample the desired number of paths with desired characteristics. Paths generated for 4 example origin-destination pairs in Ingolstadt, Provins, and St. Arnoult traffic networks are depicted in Figure 5.

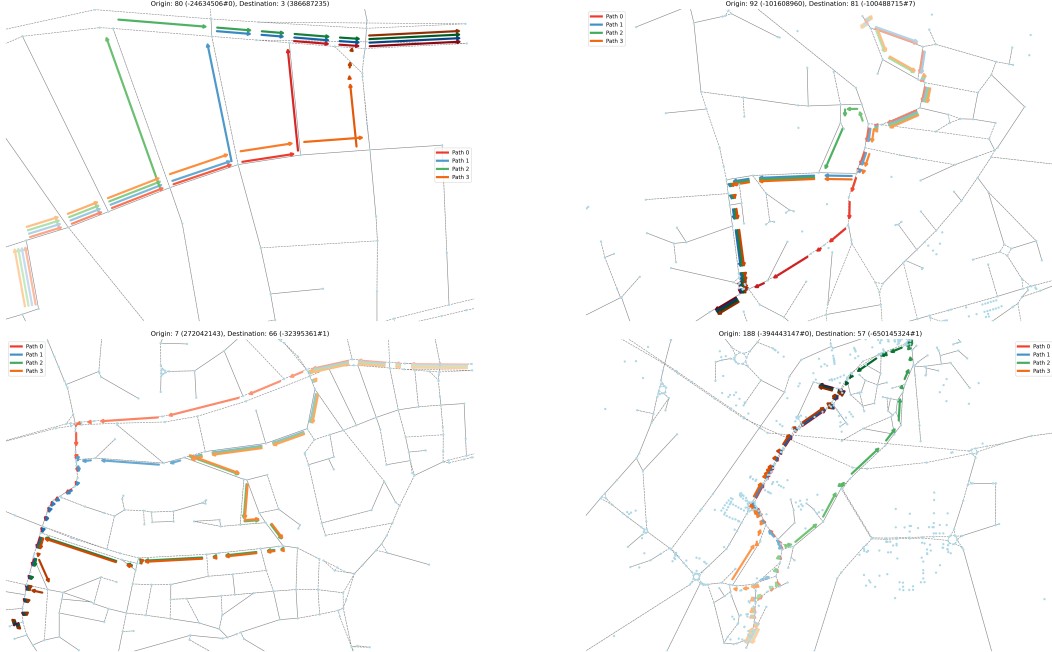

Figure 5: Routes generated for 4 selected origin-destination pairs in 3 different traffic networks used in our experiments (Ingolstadt (left), St. Arnoult (top right), Provins (bottom right)). Each shading color represents a different route.

# F Demand statistics and network layouts

For each traffic network included in URB, Table 7 shows the number of trips and the number of unique origin-destination pairs in the demand data. The network layouts are depicted in Figure 6.

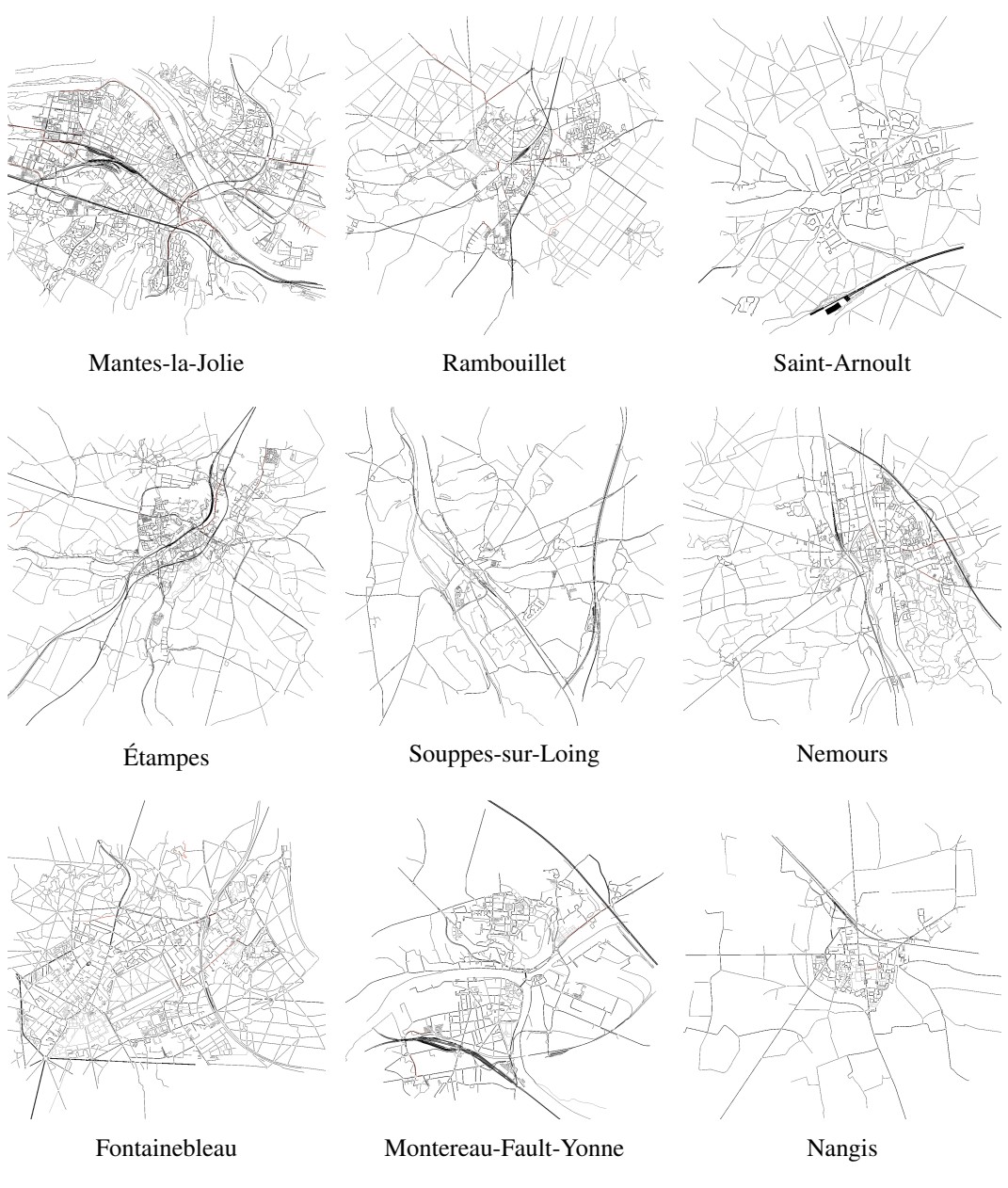

Mantes-la-Jolie     Rambouillet     Saint-Arnoult

Étampes     Souppes-sur-Loing     Nemours

Fontainebleau     Montereau-Fault-Yonne     Nangis

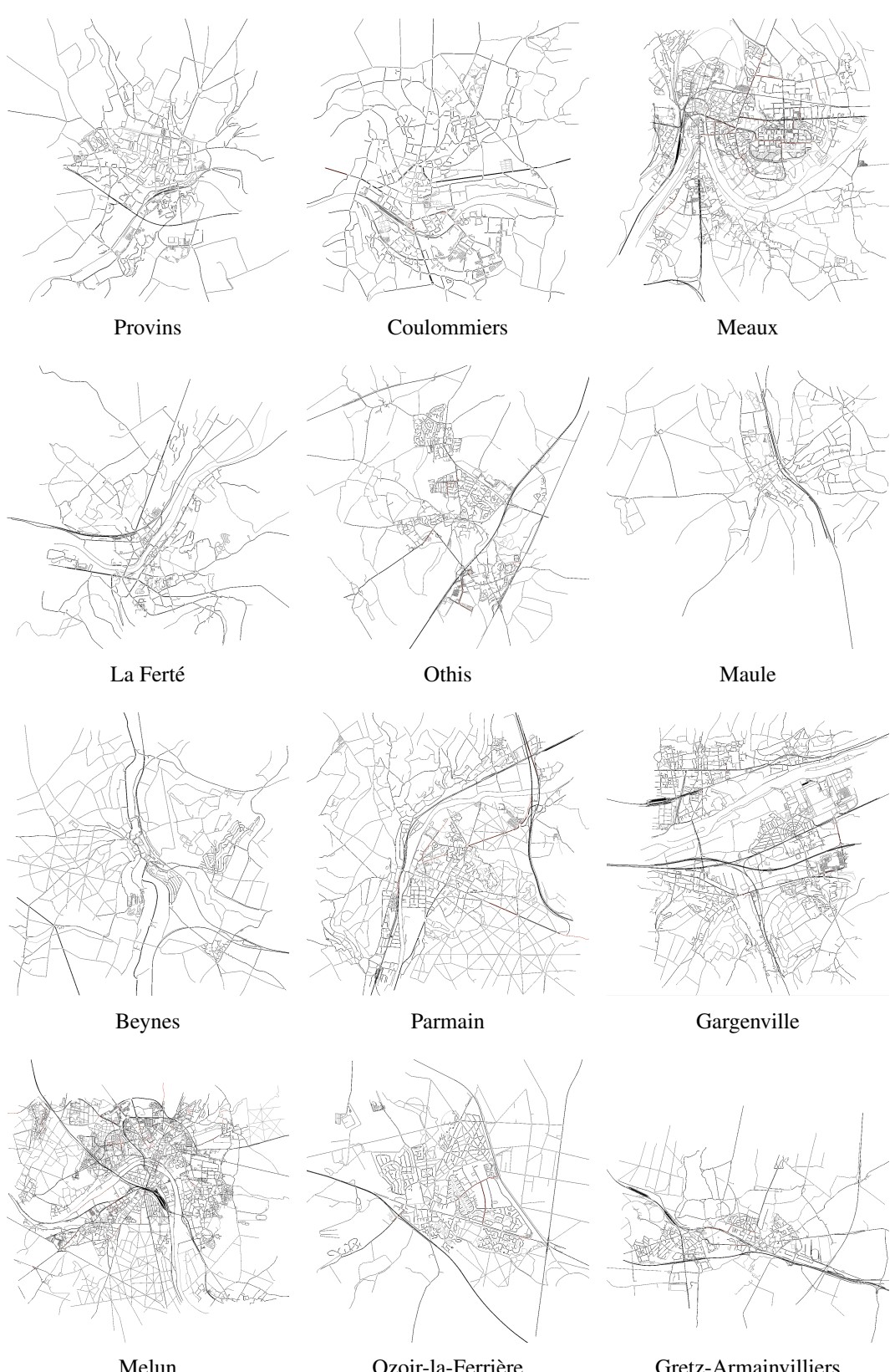

Provins      Coulommiers      Meaux

La Ferté      Othis      Maule

Beynes      Parmain      Gargenville

Melun      Ozoir-la-Ferrière      Gretz-Armainvilliers

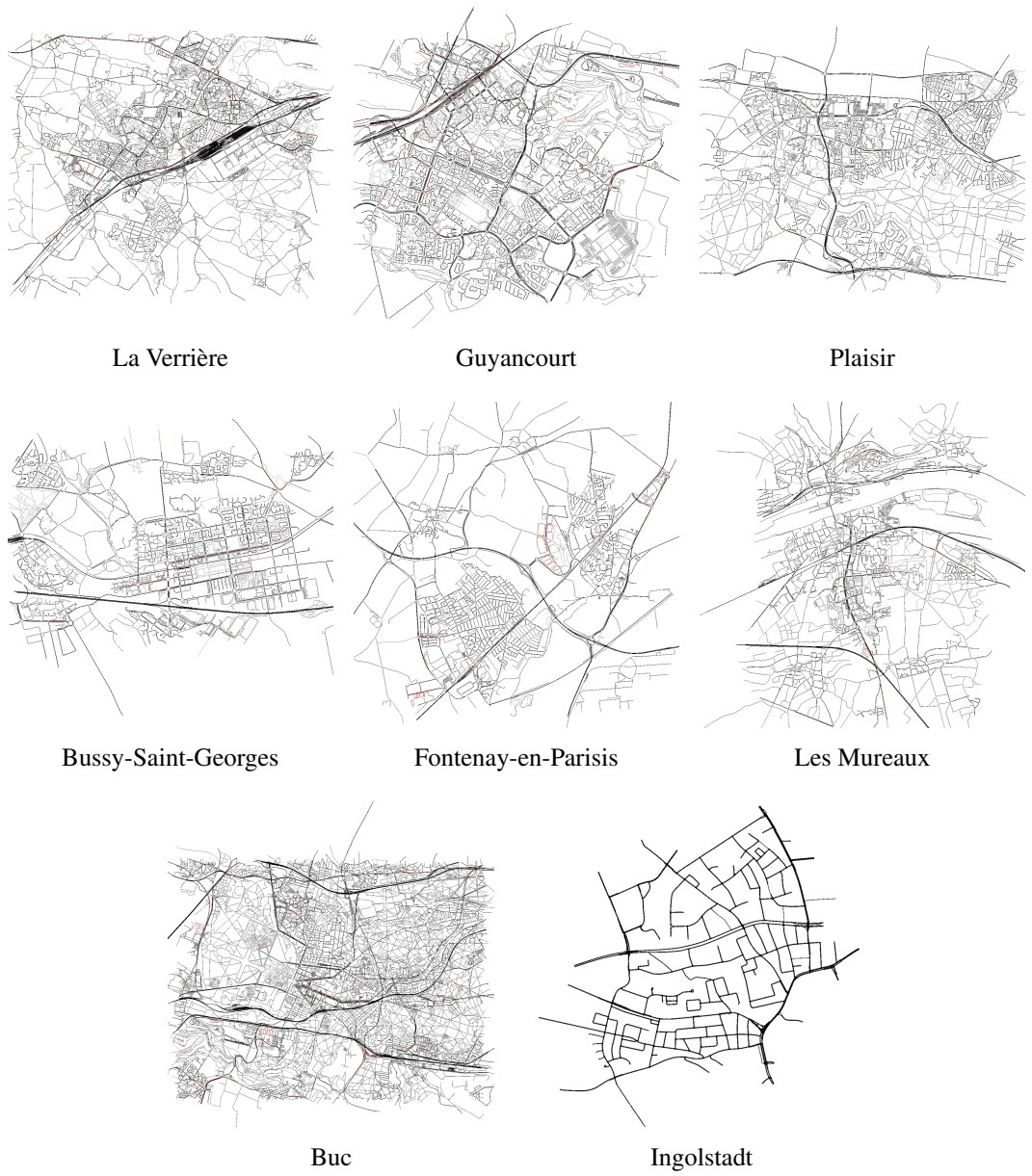

Figure 6: All 29 traffic networks included in URB. First 28 networks are from subregions of Île-de-France, and the last network, Ingolstadt, is imported from RESCO [5].

# G  Supplementary results

## G.1  Scenario 2: Full autonomy

Results reported in 4.1 are complemented with a secondary task, where we investigated what happens when *all human drivers in St. Arnoult are replaced by CAVs*. Figure 7 depicts the mean travel time changes of the CAV fleet, trained with the same algorithms and parameterizations. Similar to the first scenario, many algorithms oscillate near random baseline performance; none of the algorithms beat the All-or-Nothing solution. This suggests that the aforementioned issues may also persist in CAV-only systems, extending the scope of the identified methodological shortcomings to a greater variety of future traffic scenarios.

Table 7: The number of trips and unique origin-destinations in the demand per region.

| Region | Number of trips | Unique OD pairs |
| --- | --- | --- |
| Mantes-la-Jolie | 4271 | 4143 |
| Rambouillet | 1507 | 1450 |
| Saint-Arnoult | 222 | 215 |
| Étampes | 1159 | 1124 |
| Souppes-sur-Loing | 205 | 204 |
| Nemours | 729 | 724 |
| Fontainebleau | 1352 | 1325 |
| Montereau-Fault-Yonne | 778 | 761 |
| Nangis | 362 | 352 |
| Provins | 523 | 517 |
| Coulommiers | 549 | 542 |
| Meaux | 2637 | 2590 |
| La Ferté | 269 | 267 |
| Othis | 841 | 830 |
| Maule | 225 | 221 |
| Beynes | 234 | 229 |
| Parmain | 761 | 753 |
| Gargenville | 1000 | 960 |
| Melun | 4107 | 4054 |
| Ozoir-la-Ferrière | 643 | 633 |
| Gretz-Armainvilliers | 636 | 629 |
| La Verrière | 3521 | 3431 |
| Guyancourt | 2405 | 2352 |
| Plaisir | 1924 | 1867 |
| Bussy-Saint-Georges | 724 | 714 |
| Fontenay-en-Parisis | 2068 | 2018 |
| Les Mureaux | 2112 | 2049 |
| Buc | 6924 | 6791 |
| Ingolstadt | 1035 | 306 |

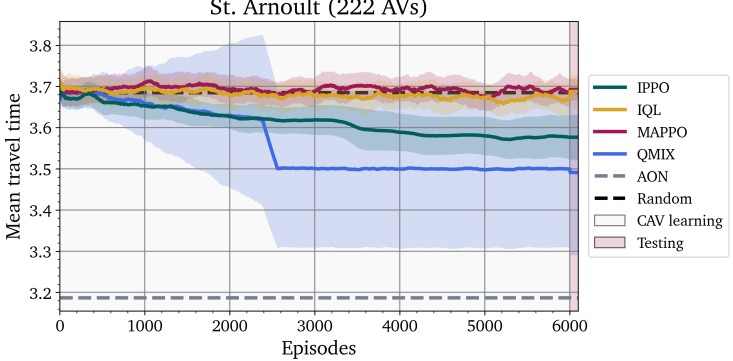

Figure 7: Mean travel times (in minutes) across episodes in St. Arnoult for Scenario 2 (Full autonomy). We report the mean CAV travel times along with 95% confidence intervals for five seeded runs of each algorithm. Smoothed using a moving average of 150 episodes. Background patches indicate phases: 6 000 and 100 episodes (days simulated) for the CAV training and policy testing, respectively. None of the methods beat the `AON` baseline, mostly oscillating around the random policy performance. QMIX exhibits the most notable learning performance, though its performance is highly inconsistent across trials (also evident in Scenario 1).

## G.2 Adjustments in algorithm implementations

Our findings (reported in Section 4) in Scenario 1 motivated us to explore modifications to existing methods in order to improve learning performance. Specifically, we propose adjustments and certain simplifications in the community implementations of IQL and IPPO, which we believe make them better suited for our problem. These changes evidently yield superior performance, now beating the benchmarks with greater consistency, and are good candidates to dominate our leaderboard. Specifically, we eliminated temporal credit assignment mechanisms (the bootstrapping term in DQN loss and critic estimation in PPO advantage) and used only local experiences in learning. Both implementations improve the performance of their predecessors, as reported in Table 8, with 100% WR in St. Arnoult and Provins by IQL, indicating that the CAVs consistently achieved shorter mean travel times than human drivers after training. These alternative implementations are also included in our repository (see Section C.1).

Table 8: Results of the independent learning algorithms with their adjusted implementations (*), and naive baselines (taken from Table 1) in Scenario 1. Each value reports the mean and standard deviation across five seeded runs, except for the baselines. Each RL experiment followed three subsequent phases: Human stabilization (200 episodes), CAV learning (2 000 episodes), and policy testing (100 episodes). For each network and metric, the best RL performance is underlined and the best result overall is highlighted in **bold**. Metrics used and $t^{\text{pre}}$ for each network are the same as in Table 1. The results reported here evidence the improved performance achieved by the new implementations. Notably, both IPPO* and IQL* surpassed their general-use implementation versions. In St. Arnoult and Provins, IQL*-trained CAVs now consistently perform better compared to the pre-CAV travel times of humans.

|  |  | $t_{\text{CAV}}$ | $c_{\text{CAV}}$ | WR |
|---|---|---|---|---|
| ST. ARNOULT | IPPO* | 3.31 (0.05) | **0.80** (0.06) | 0% |
|  | IPPO | 3.33 (0.013) | 1.38 (0.034) | 0% |
|  | IQL* | 3.02 (0.011) | 1.41 (0.0) | **100%** |
|  | IQL | 3.53 (0.104) | 1.44 (0.004) | 0% |
|  | AON | **3.01** | 1.21 | **100%** |
|  | RANDOM | 3.58 | 1.36 | 0% |
| PROVINS | IPPO* | 2.88 (0.013) | **0.64** (0.09) | 0% |
|  | IPPO | 2.98 (0.04) | 1.05 (0.356) | 0% |
|  | IQL* | **2.76** (0.008) | 1.19 (0.186) | **100%** |
|  | IQL | 3.01 (0.027) | 2.12 (0.183) | 0% |
|  | AON | **2.76** | 0.99 | **100%** |
|  | RANDOM | 3.04 | 0.95 | 0% |
| INGOLSTADT | IPPO* | 4.56 (0.025) | 1.86 (0.535) | 0% |
|  | IPPO | 4.71 (0.03) | 3.19 (0.495) | 0% |
|  | IQL* | **4.22** (0.015) | 2.01 (0.27) | 0% |
|  | IQL | 4.81 (0.024) | 3.44 (0.562) | 0% |
|  | AON | 4.37 | **0.24** | 0% |
|  | RANDOM | 4.81 | 1.74 | 0% |

## G.3 Demonstrative experiments

To further showcase URB's flexibility and capabilities, we complement the results reported in this paper with three demonstrative experiments:

- **Selfish CAVs versus adapting humans in Nangis**: In **Nangis**, what happens when the **40%** of the drivers convert into **selfish** CAVs, who thereafter learn routing strategies with the **IPPO** algorithm while the remaining humans are simultaneously **adapting** to these changes?
- **Malicious CAVs in Nemours**: In **Nemours**, what happens when the **40%** of the drivers convert into **malicious** CAVs, who learn to maximize negative impact for humans, with the **QMIX** algorithm?

- **Altruistic CAVs in Gretz-Armainvilliers**: In **Gretz-Armainvilliers**, what happens when the **40%** of the drivers convert into **altruistic** CAVs, who aim to improve overall traffic efficiency, with the **VDN** algorithm, while the remaining humans are simultaneously **adapting** to these changes?

The experiment results, conducted with parameterization shared in our result data repository as described in Appendix B, are provided in Figure 8.

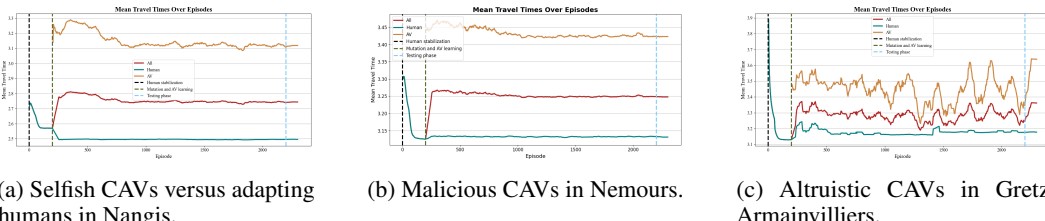

(a) Selfish CAVs versus adapting humans in Nangis.

(b) Malicious CAVs in Nemours.

(c) Altruistic CAVs in Gretz-Armainvilliers.

Figure 8: Mean travel times (in minutes) of humans and CAVs over the episodes in demonstrative experiments.

## G.4 Additional plots: Travel times across episodes

By default, at the end of each `URB` experiment, several plots are generated using RouteRL's built-in plotting functions. These visualizations can be a starting point for analyses and are excellent for early detection of potential issues. In this section, we share the plots for travel time changes produced for the experiments reported in this study, including all repetitions.

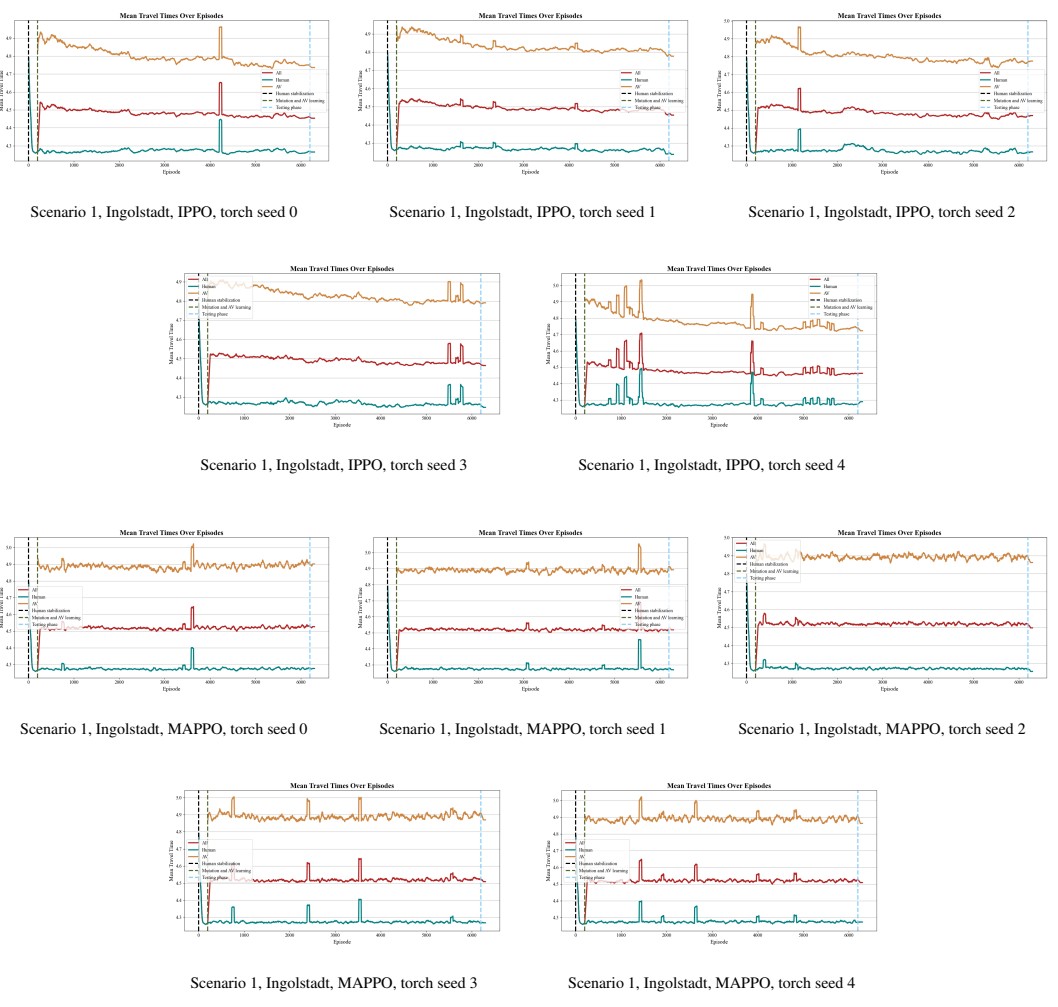

Scenario 1, Ingolstadt, IPPO, torch seed 0

Scenario 1, Ingolstadt, IPPO, torch seed 1

Scenario 1, Ingolstadt, IPPO, torch seed 2

Scenario 1, Ingolstadt, IPPO, torch seed 3

Scenario 1, Ingolstadt, IPPO, torch seed 4

Scenario 1, Ingolstadt, MAPPO, torch seed 0

Scenario 1, Ingolstadt, MAPPO, torch seed 1

Scenario 1, Ingolstadt, MAPPO, torch seed 2

Scenario 1, Ingolstadt, MAPPO, torch seed 3

Scenario 1, Ingolstadt, MAPPO, torch seed 4

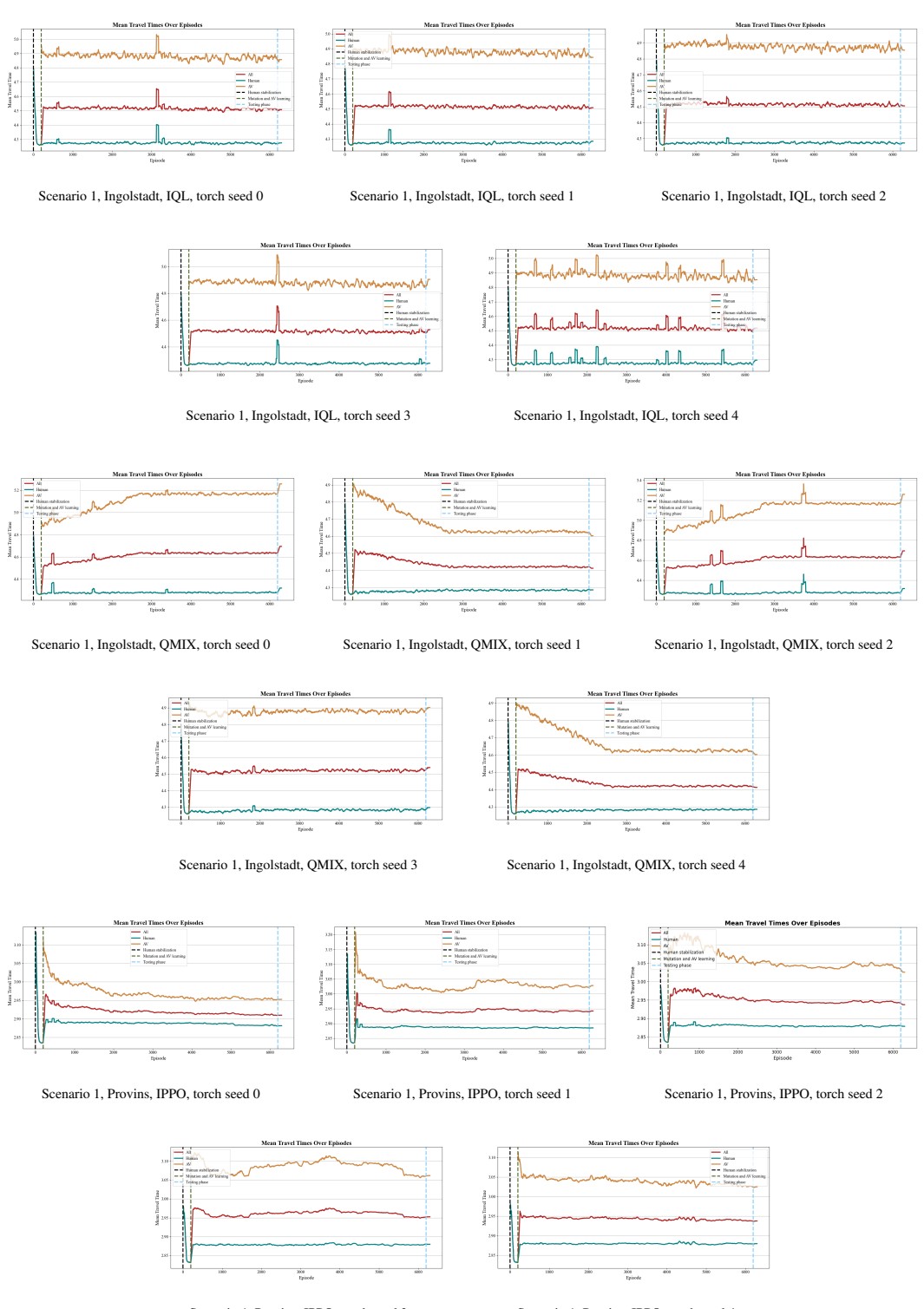

Scenario 1, Ingolstadt, IQL, torch seed 0     Scenario 1, Ingolstadt, IQL, torch seed 1     Scenario 1, Ingolstadt, IQL, torch seed 2

Scenario 1, Ingolstadt, IQL, torch seed 3     Scenario 1, Ingolstadt, IQL, torch seed 4

Scenario 1, Ingolstadt, QMIX, torch seed 0     Scenario 1, Ingolstadt, QMIX, torch seed 1     Scenario 1, Ingolstadt, QMIX, torch seed 2

Scenario 1, Ingolstadt, QMIX, torch seed 3     Scenario 1, Ingolstadt, QMIX, torch seed 4

Scenario 1, Provins, IPPO, torch seed 0     Scenario 1, Provins, IPPO, torch seed 1     Scenario 1, Provins, IPPO, torch seed 2

Scenario 1, Provins, IPPO, torch seed 3     Scenario 1, Provins, IPPO, torch seed 4

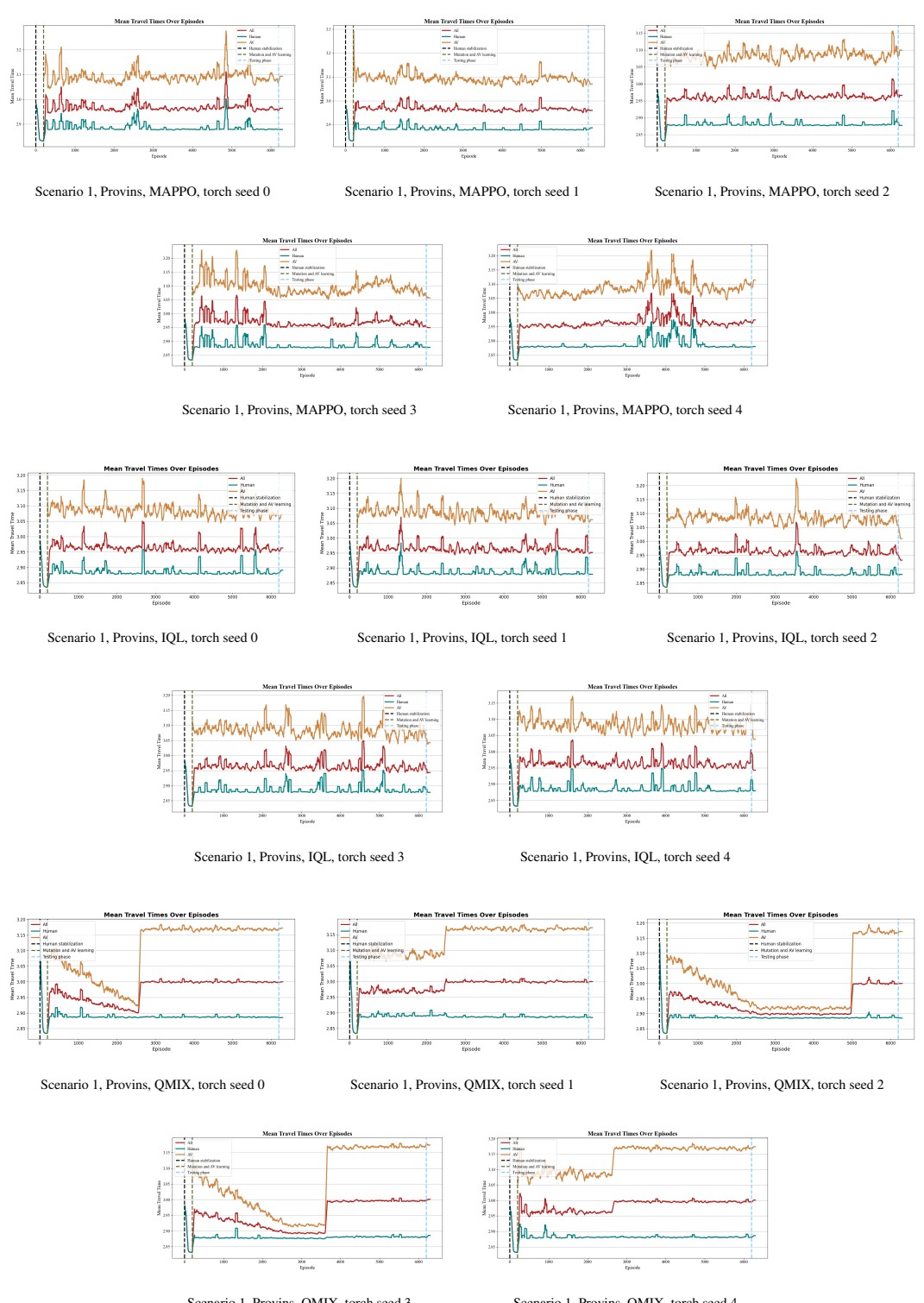

Scenario 1, Provins, MAPPO, torch seed 0    Scenario 1, Provins, MAPPO, torch seed 1    Scenario 1, Provins, MAPPO, torch seed 2

Scenario 1, Provins, MAPPO, torch seed 3    Scenario 1, Provins, MAPPO, torch seed 4

Scenario 1, Provins, IQL, torch seed 0    Scenario 1, Provins, IQL, torch seed 1    Scenario 1, Provins, IQL, torch seed 2

Scenario 1, Provins, IQL, torch seed 3    Scenario 1, Provins, IQL, torch seed 4

Scenario 1, Provins, QMIX, torch seed 0    Scenario 1, Provins, QMIX, torch seed 1    Scenario 1, Provins, QMIX, torch seed 2

Scenario 1, Provins, QMIX, torch seed 3    Scenario 1, Provins, QMIX, torch seed 4

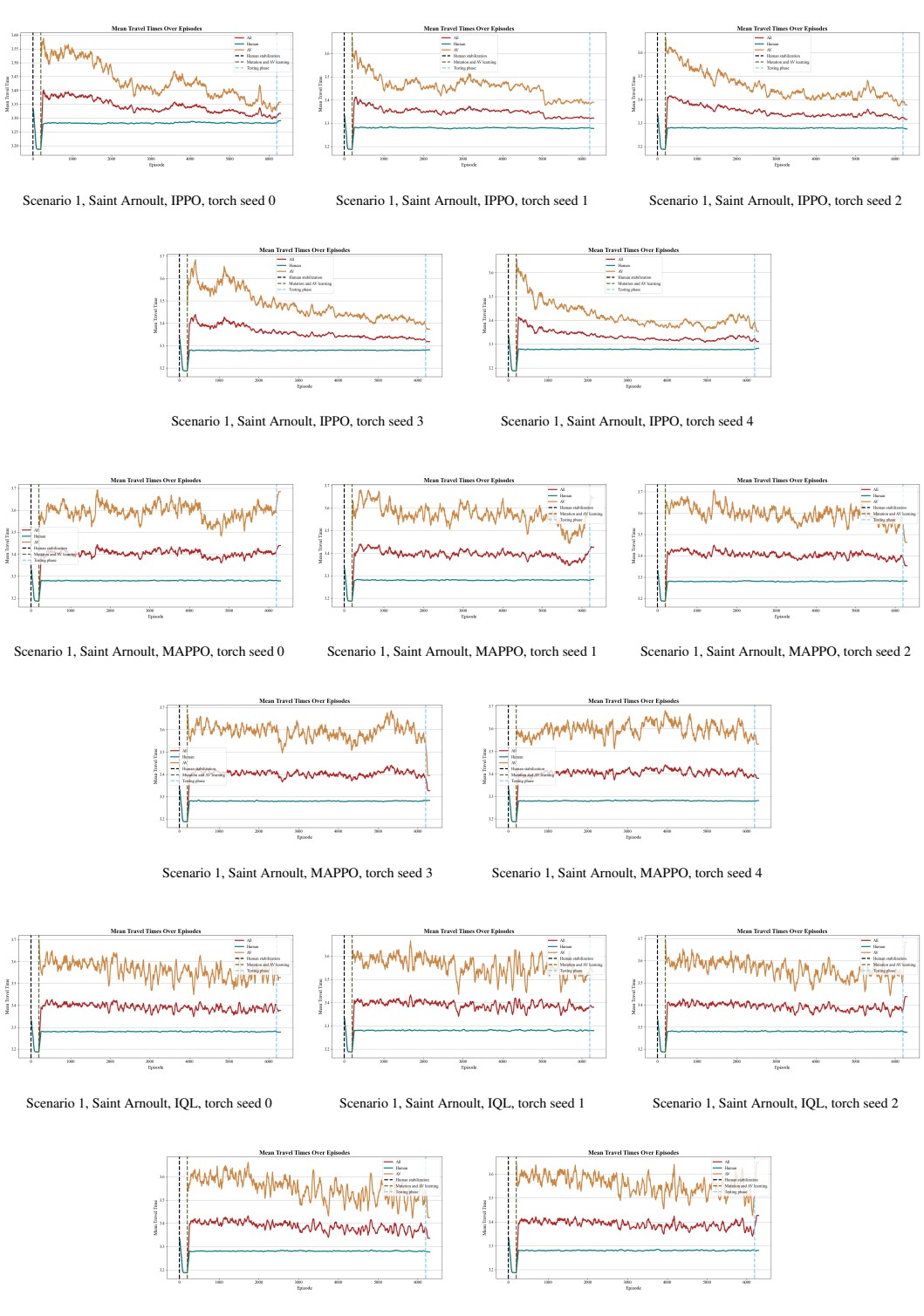

Scenario 1, Saint Arnoult, IPPO, torch seed 0     Scenario 1, Saint Arnoult, IPPO, torch seed 1     Scenario 1, Saint Arnoult, IPPO, torch seed 2

Scenario 1, Saint Arnoult, IPPO, torch seed 3     Scenario 1, Saint Arnoult, IPPO, torch seed 4

Scenario 1, Saint Arnoult, MAPPO, torch seed 0     Scenario 1, Saint Arnoult, MAPPO, torch seed 1     Scenario 1, Saint Arnoult, MAPPO, torch seed 2

Scenario 1, Saint Arnoult, MAPPO, torch seed 3     Scenario 1, Saint Arnoult, MAPPO, torch seed 4

Scenario 1, Saint Arnoult, IQL, torch seed 0     Scenario 1, Saint Arnoult, IQL, torch seed 1     Scenario 1, Saint Arnoult, IQL, torch seed 2

Scenario 1, Saint Arnoult, IQL, torch seed 3     Scenario 1, Saint Arnoult, IQL, torch seed 4

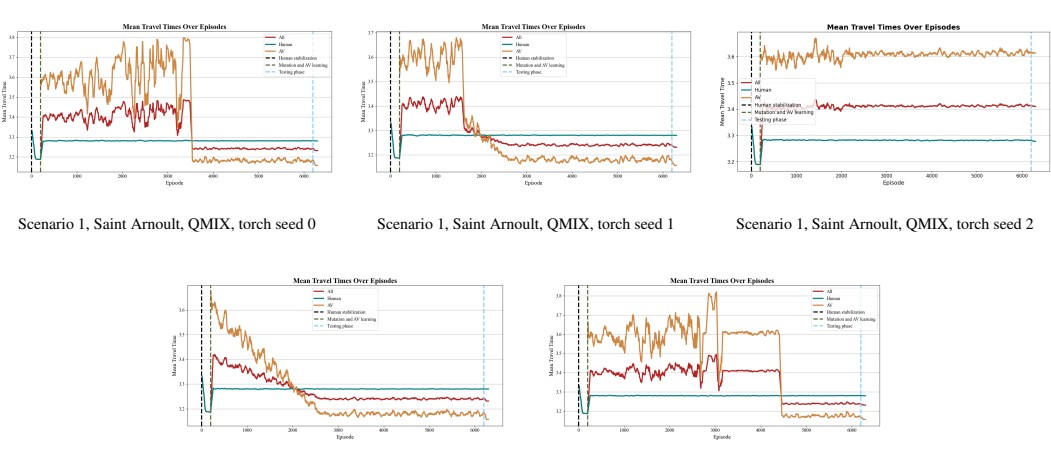

Scenario 1, Saint Arnoult, QMIX, torch seed 0      Scenario 1, Saint Arnoult, QMIX, torch seed 1      Scenario 1, Saint Arnoult, QMIX, torch seed 2

Scenario 1, Saint Arnoult, QMIX, torch seed 3      Scenario 1, Saint Arnoult, QMIX, torch seed 4

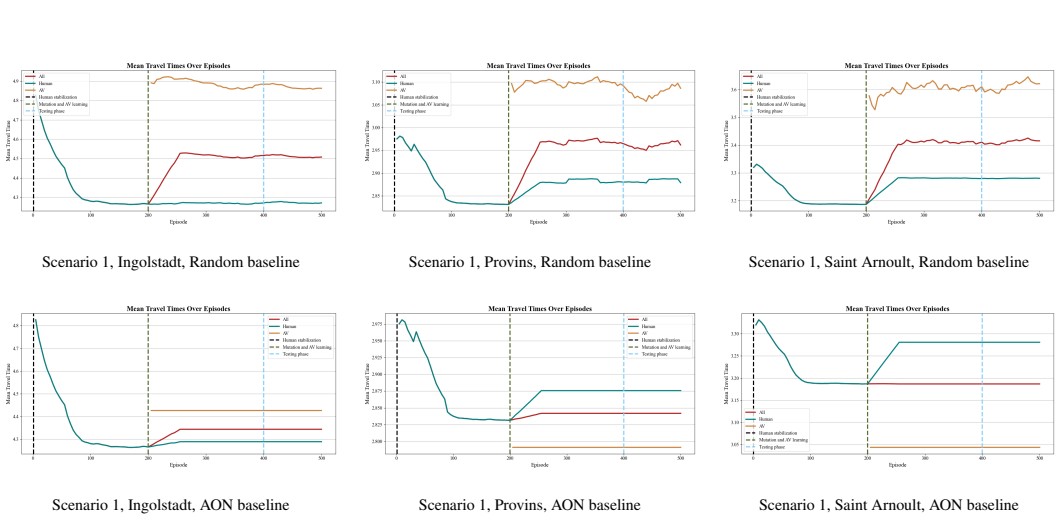

Scenario 1, Ingolstadt, Random baseline      Scenario 1, Provins, Random baseline      Scenario 1, Saint Arnoult, Random baseline

Scenario 1, Ingolstadt, AON baseline      Scenario 1, Provins, AON baseline      Scenario 1, Saint Arnoult, AON baseline

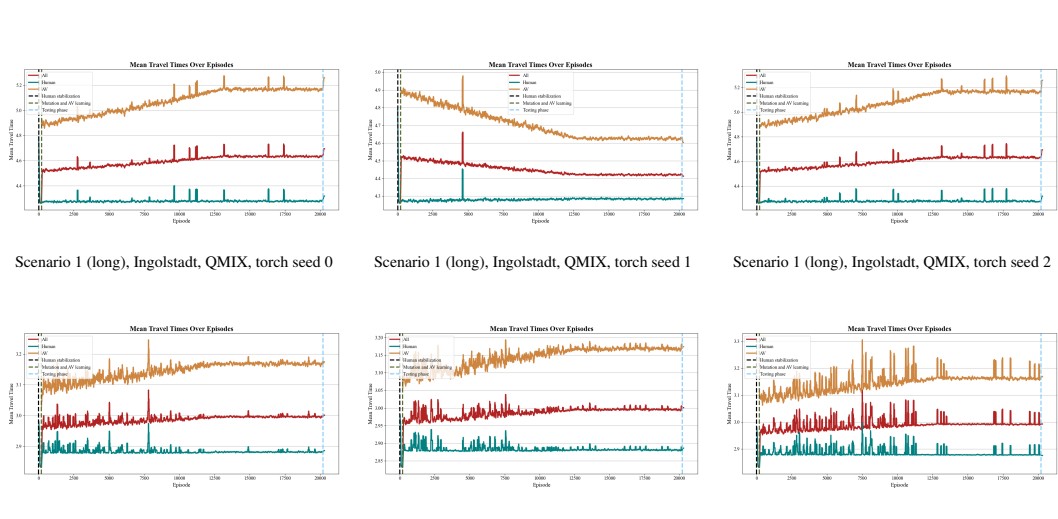

Scenario 1 (long), Ingolstadt, QMIX, torch seed 0      Scenario 1 (long), Ingolstadt, QMIX, torch seed 1      Scenario 1 (long), Ingolstadt, QMIX, torch seed 2

Scenario 1 (long), Provins, QMIX, torch seed 0      Scenario 1 (long), Provins, QMIX, torch seed 1      Scenario 1 (long), Provins, QMIX, torch seed 2

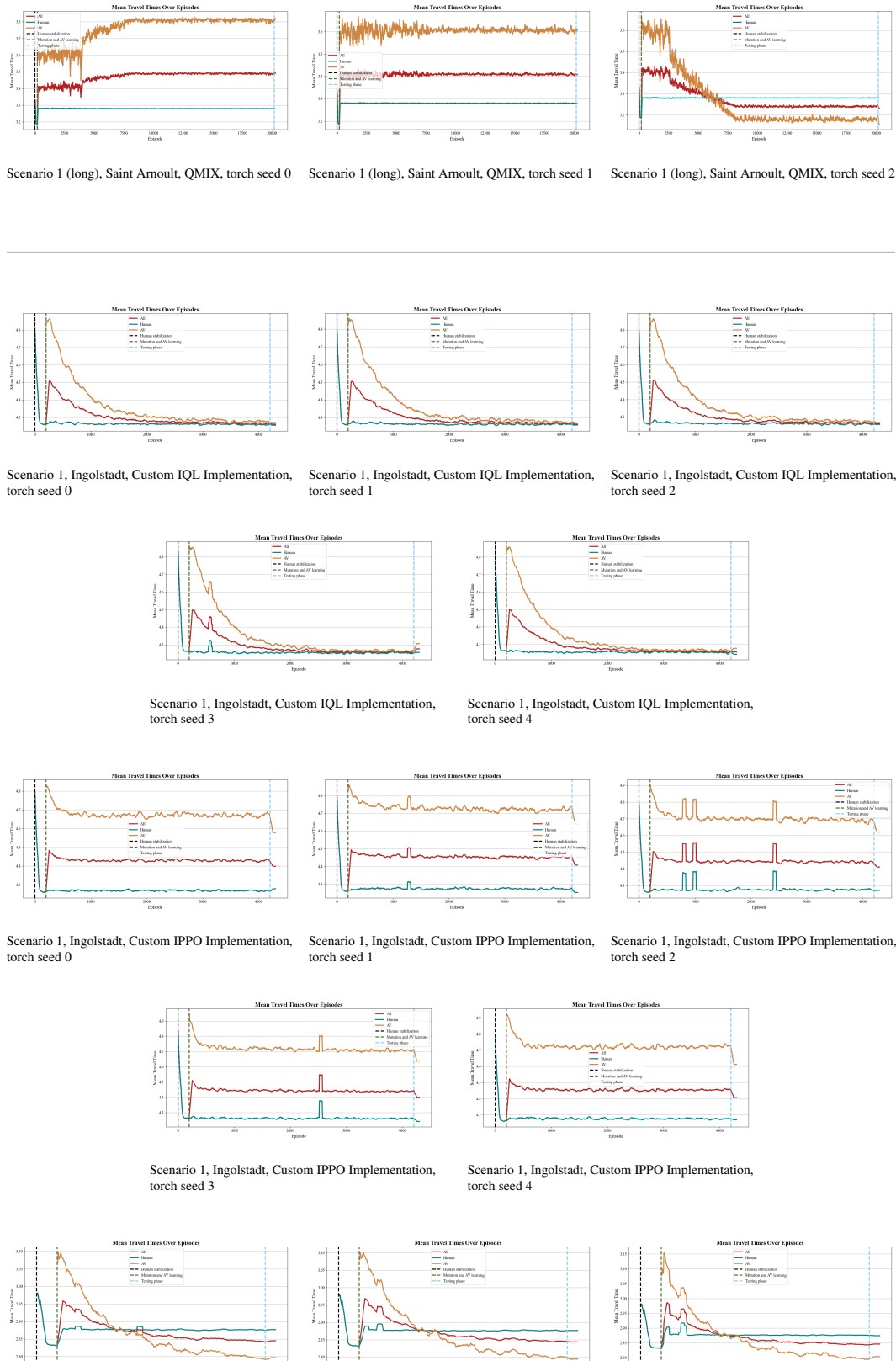

Scenario 1 (long), Saint Arnoult, QMIX, torch seed 0  Scenario 1 (long), Saint Arnoult, QMIX, torch seed 1  Scenario 1 (long), Saint Arnoult, QMIX, torch seed 2

Scenario 1, Ingolstadt, Custom IQL Implementation, torch seed 0

Scenario 1, Ingolstadt, Custom IQL Implementation, torch seed 1

Scenario 1, Ingolstadt, Custom IQL Implementation, torch seed 2

Scenario 1, Ingolstadt, Custom IQL Implementation, torch seed 3

Scenario 1, Ingolstadt, Custom IQL Implementation, torch seed 4

Scenario 1, Ingolstadt, Custom IPPO Implementation, torch seed 0

Scenario 1, Ingolstadt, Custom IPPO Implementation, torch seed 1

Scenario 1, Ingolstadt, Custom IPPO Implementation, torch seed 2

Scenario 1, Ingolstadt, Custom IPPO Implementation, torch seed 3

Scenario 1, Ingolstadt, Custom IPPO Implementation, torch seed 4

Scenario 1, Provins, Custom IQL Implementation, torch seed 0

Scenario 1, Provins, Custom IQL Implementation, torch seed 1

Scenario 1, Provins, Custom IQL Implementation, torch seed 2

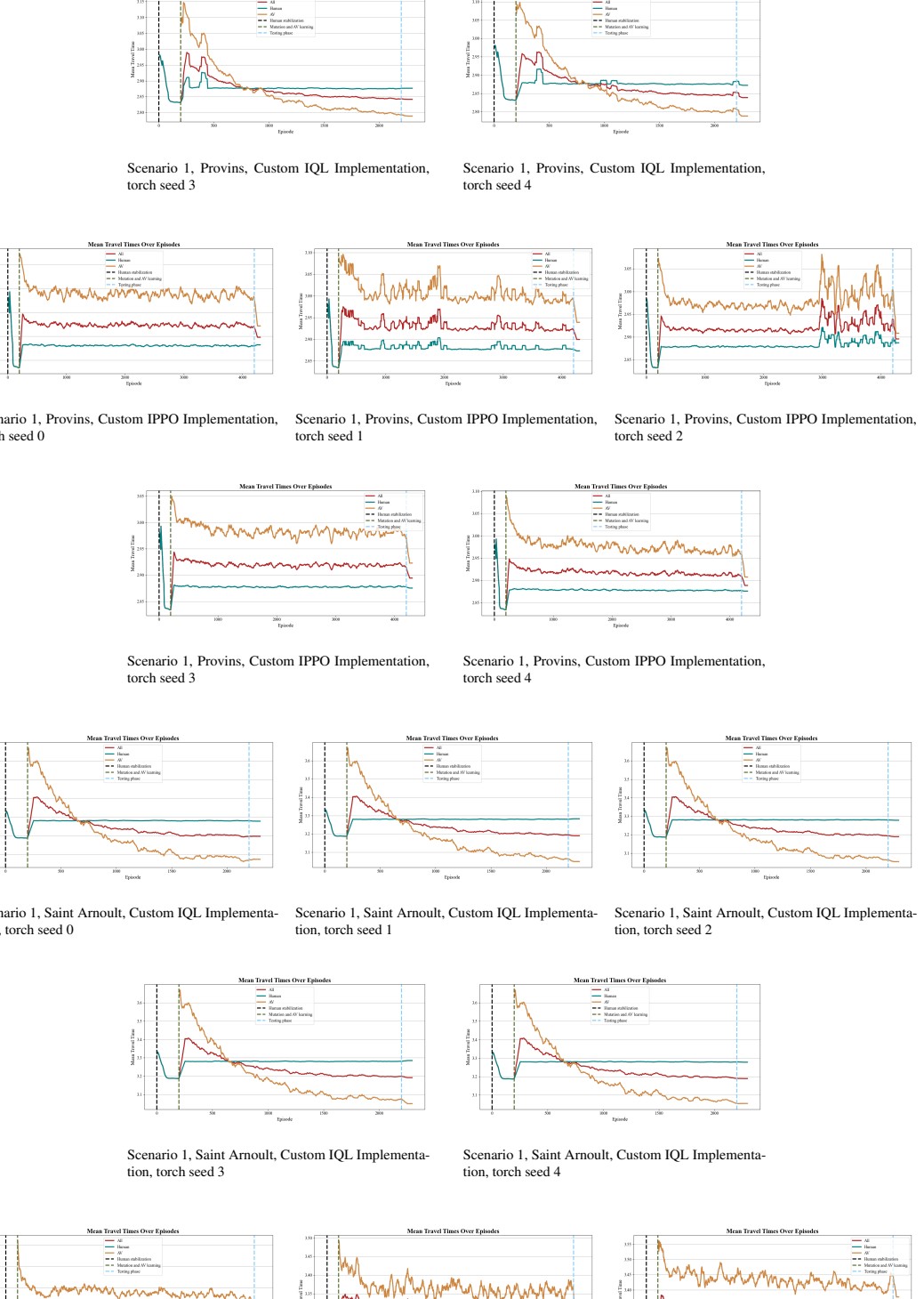

Scenario 1, Provins, Custom IQL Implementation, torch seed 3

Scenario 1, Provins, Custom IQL Implementation, torch seed 4

Scenario 1, Provins, Custom IPPO Implementation, torch seed 0

Scenario 1, Provins, Custom IPPO Implementation, torch seed 1

Scenario 1, Provins, Custom IPPO Implementation, torch seed 2

Scenario 1, Provins, Custom IPPO Implementation, torch seed 3

Scenario 1, Provins, Custom IPPO Implementation, torch seed 4

Scenario 1, Saint Arnoult, Custom IQL Implementation, torch seed 0

Scenario 1, Saint Arnoult, Custom IQL Implementation, torch seed 1

Scenario 1, Saint Arnoult, Custom IQL Implementation, torch seed 2

Scenario 1, Saint Arnoult, Custom IQL Implementation, torch seed 3

Scenario 1, Saint Arnoult, Custom IQL Implementation, torch seed 4

Scenario 1, Saint Arnoult, Custom IPPO Implementation, torch seed 0

Scenario 1, Saint Arnoult, Custom IPPO Implementation, torch seed 1

Scenario 1, Saint Arnoult, Custom IPPO Implementation, torch seed 2

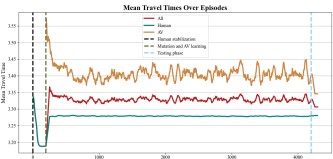

Scenario 1, Saint Arnoult, Custom IPPO Implementation, torch seed 3

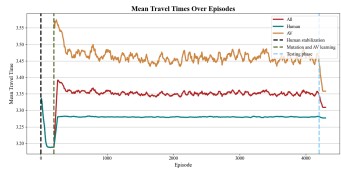

Scenario 1, Saint Arnoult, Custom IPPO Implementation, torch seed 4

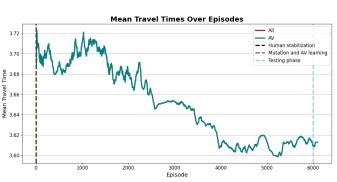

Scenario 2, Saint Arnoult, IPPO, torch seed 0

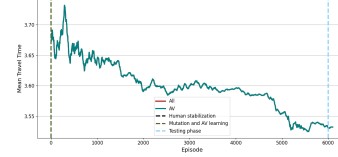

Scenario 2, Saint Arnoult, IPPO, torch seed 1

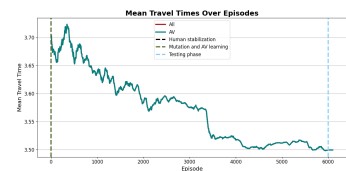

Scenario 2, Saint Arnoult, IPPO, torch seed 2

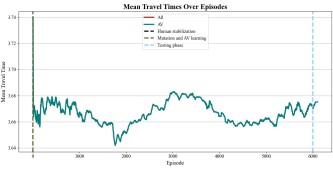

Scenario 2, Saint Arnoult, IPPO, torch seed 3

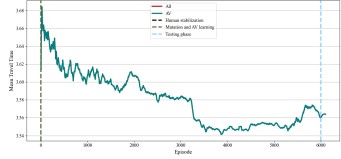

Scenario 2, Saint Arnoult, IPPO, torch seed 4

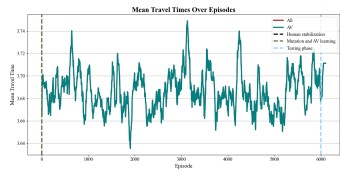

Scenario 2, Saint Arnoult, MAPPO, torch seed 0

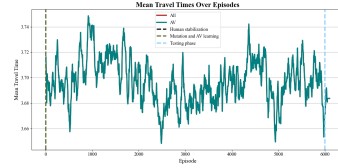

Scenario 2, Saint Arnoult, MAPPO, torch seed 1

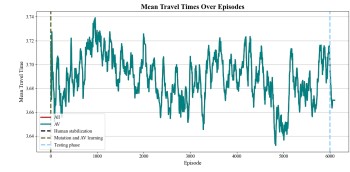

Scenario 2, Saint Arnoult, MAPPO, torch seed 2

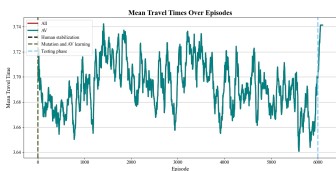

Scenario 2, Saint Arnoult, MAPPO, torch seed 3

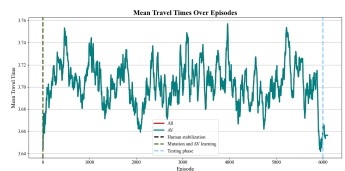

Scenario 2, Saint Arnoult, MAPPO, torch seed 4

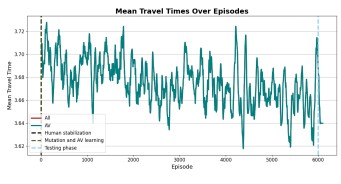

Scenario 2, Saint Arnoult, IQL, torch seed 0

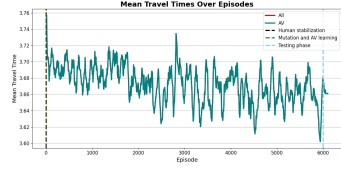

Scenario 2, Saint Arnoult, IQL, torch seed 1

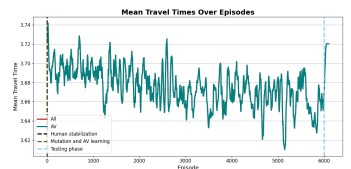

Scenario 2, Saint Arnoult, IQL, torch seed 2

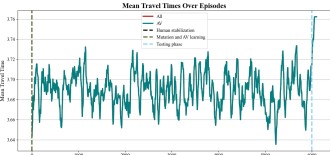
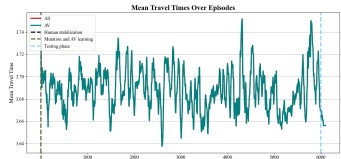

Scenario 2, Saint Arnoult, IQL, torch seed 3    Scenario 2, Saint Arnoult, IQL, torch seed 4

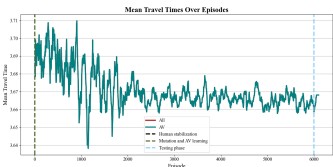
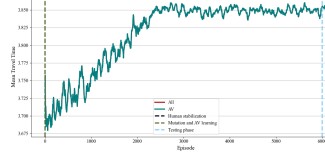
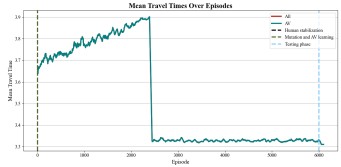

Scenario 2, Saint Arnoult, QMIX, torch seed 0    Scenario 2, Saint Arnoult, QMIX, torch seed 1    Scenario 2, Saint Arnoult, QMIX, torch seed 2

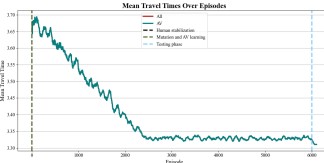
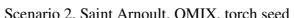
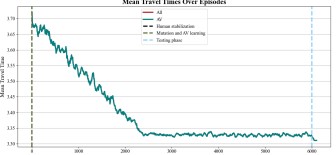

Scenario 2, Saint Arnoult, QMIX, torch seed 3    Scenario 2, Saint Arnoult, QMIX, torch seed 4

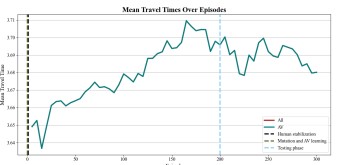
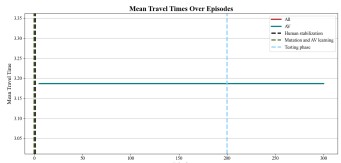

Scenario 2, Saint Arnoult, Random baseline    Scenario 2, Saint Arnoult, AON baseline

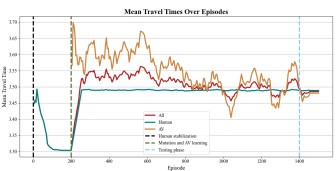
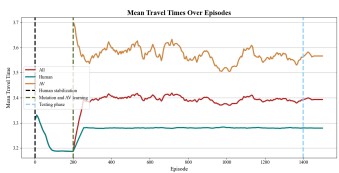

Scenario 1, sensitivity analysis: Half demand in St. Arnoult, IQL    Scenario 1, sensitivity analysis: Normal demand in St. Arnoult, IQL

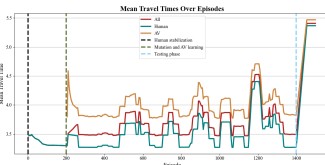
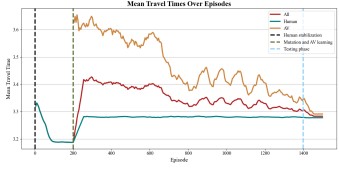

Scenario 1, sensitivity analysis: Double demand in St. Arnoult, IQL    Scenario 1, sensitivity analysis: Normal demand in St. Arnoult, IQL with global observations

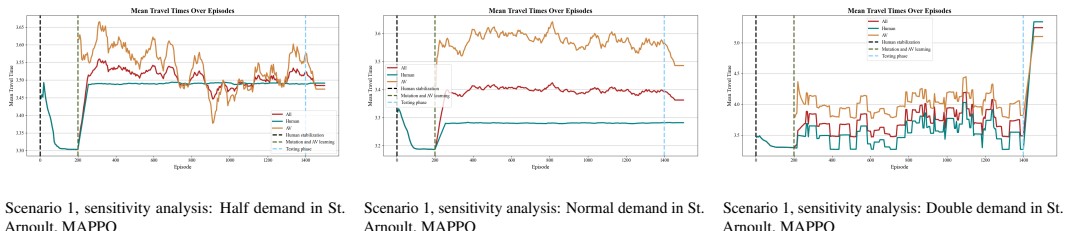

Scenario 1, sensitivity analysis: Half demand in St. Arnoult, MAPPO

Scenario 1, sensitivity analysis: Normal demand in St. Arnoult, MAPPO

Scenario 1, sensitivity analysis: Double demand in St. Arnoult, MAPPO

Figure 9: Travel times across episodes for all experiments reported in this study.

