# OpenReview forum: "URB - Urban Routing Benchmark for RL-equipped Connected Autonomous Vehicles"
_NeurIPS.cc/2025/Datasets_and_Benchmarks_Track — NeurIPS 2025 Datasets and Benchmarks Track poster_

### Official Review · Reviewer_2EFp · 2025-06-28

**Rating:** 4
**Confidence:** 4

**Summary:**

This paper introduces URB, a comprehensive benchmark for evaluating multi-agent reinforcement learning (MARL) algorithms in urban routing scenarios involving Connected Autonomous Vehicles (CAVs). URB includes 29 real-world traffic networks with realistic demand data, supports various routing objectives and behaviors, and integrates four state-of-the-art MARL algorithms and three baseline methods.

**Dataset Code Accessibility:**

Yes

**Dataset Code Comments:**

The authors provide the corresponding code and datasets in the github repository and give detailed instructions.

**Ethical Considerations:**

No, there are no or only very minor ethics concerns

**Final Justification:**

Thanks to the authors for their response. I still have some concerns about the performance of the methods included in the benchmark, but the authors also provided some explanations, so I maintain my borderline acceptance rating.

**Limitations Weaknesses:**

i) Although the paper evaluates several mainstream MARL algorithms, none consistently outperform human or simple baselines (e.g., All-or-Nothing) across most scenarios. Moreover, the study focuses solely on benchmarking without proposing any novel or improved methods.

ii) The experiments assume that human strategies are stable before CAV deployment and that CAVs make a single, static routing decision per episode. This overlooks key real-world complexities such as dynamic traffic patterns, interactive learning, and real-time route adjustments.

**Strengths Contributions:**

i) URB integrates 29 real-world urban traffic networks along with corresponding demand data, covering road networks from small towns to large cities. It offers a high-fidelity and scalable simulation environment for CAV routing research.

ii) The authors formalize the urban CAV routing problem as an Agent-Environment Cycle (AEC) game and support various agent behaviors (e.g., selfish, altruistic, malicious). The platform allows flexible customization of CAV ratios, reward functions, route choice sets, and more, making it applicable to a wide range of research scenarios.

iii) By systematically evaluating four mainstream MARL algorithms, the paper shows that most current methods fail to outperform human baselines under realistic conditions. This provides strong empirical evidence and a clear challenge to the research community, motivating further algorithmic improvements.

---

> ### Author Rebuttal · Authors · 2025-07-31
>
> Thank you for your thorough review and valuable assessments. As authors, we have carefully considered both your positive comments and critical observations. We are pleased to see that URB's variety, flexibility, and functionalities are accurately identified and described in your review. Below, we share our responses to the weaknesses you have identified.
>
> ---
>
> > [W1a] Although the paper evaluates several mainstream MARL algorithms, none consistently outperform human or simple baselines (e.g., All-or-Nothing) across most scenarios.
>
> We agree, the reported findings make the algorithmic limitations evident. These results, which are generated using community-driven off-the-shelf implementations of SOTA algorithms, highlight the need for methodological advancements in MARL research to address large-scale and dynamic problems. Moreover, we observe that the effectiveness of a method largely depends on the problem scale (in our context, network size and demand level), and the quality of a solution is difficult to replicate across repetitions. This highlights the practical shortcomings of these methods, evidently lacking robustness and reliability in our settings. By reporting these results, we aim to encourage the community to pursue methodological innovations, which can be tested and validated using our benchmark.
>
>
>
> > [W1b] The study focuses solely on benchmarking without proposing any novel or improved methods.
>
> We would like to clarify that the goal of this study is to formalize a problem and introduce a benchmarking framework for evaluating solutions on a common ground. We report the performance of SOTA MARL solutions against naive baselines, using our proposed performance indicators. To maintain a focused narrative, we deliberately leave methodological innovations out of scope. Instead, we use this opportunity to encourage community contributions in this direction and present our framework as a foundation for such future efforts.
>
>
> > [W2] The experiments assume that human strategies are stable before CAV deployment and that CAVs make a single, static routing decision per episode. This overlooks key real-world complexities such as dynamic traffic patterns, interactive learning, and real-time route adjustments.
>
> To the best of our knowledge, our setting is one of the most complex and realistic transport system representations used in the ML community. Nonetheless, we recognize the realism gap due to the lack of complexities you mentioned. While our simulation framework RouteRL [1] allows us to include many of the standard transportation engineering models and techniques (including those that you mentioned), at this stage, we deliberately made certain simplifications to disentangle the algorithmic performance from the dynamics of the environment. Already in such a setting, most of the MARL algorithms failed (as we reported), which calls for a competitive leaderboard (triggered and maintained within URB). When the community identifies the algorithms that consistently address the simplified versions of the problem, the tasks can become increasingly complex and can be intuitively defined with different parameterizations using URB, and can be simulated in RouteRL.
>
> ---
> [1] Akman, A. O., Psarou, A., Gorczyca, Ł., Varga, Z. G., Jamróz, G., & Kucharski, R. (2025). RouteRL: Multi-agent reinforcement learning framework for urban route choice with autonomous vehicles. arXiv preprint arXiv:2502.20065.

---

> > ### Comment · Reviewer_2EFp · 2025-08-07
> >
> > I appreciate the author's response and I choose to maintain my rating.

---

> ### Author Response · Authors · 2025-08-06
>
> We would like to draw your attention to the discussion with reviewer **3LBj**, who raised similar concerns about the performance of RL algorithms (your [W1]). In our reply, we present previously unreported experiments in which we validated some hypotheses about why standard MARL (out-of-the-box TorchRL implementations) failed. Consequently, we identified improvements to IQL and IPPO implementations, which now beat analytical and human benchmarks with more consistency, as now reported in Table 3 of the reply to reviewer **3LBj**.

---

### Official Review · Reviewer_bjfp · 2025-06-29

**Rating:** 4
**Confidence:** 4

**Summary:**

The paper presents URB (Urban Routing Benchmark), an open‐source, large‐scale framework for evaluating multi‐agent reinforcement‐learning routing policies for connected autonomous vehicles, bundling 29 real‐world traffic networks with peak‐hour demand data, a configurable SUMO‐based simulation pipeline, JanuX route generation, and task/config templates; it includes reference implementations of four state‐of‐the‐art MARL algorithms (IQL, IPPO, MAPPO, QMIX) and three baselines, all released under permissive licenses with full hyperparameter, seed, and hardware documentation (plus a one‐click Code‐Ocean capsule), and its experiments on three cities demonstrate that current MARL methods rarely outperform human routing, underscoring both the benchmark’s utility and the need for further algorithmic advances.

**Dataset Code Accessibility:**

Yes

**Dataset Code Comments:**

URB’s dataset—29 real-world traffic networks with peak-hour OD matrices—is released under MIT on GitHub and Kaggle with full documentation, and its reference implementations for IQL, IPPO, MAPPO, QMIX and baseline policies are open-sourced under MIT, complete with hyperparameters, seeds, simulation scripts, and a one-click Code-Ocean capsule for immediate reproducibility.

**Ethical Considerations:**

No, there are no or only very minor ethics concerns

**Limitations Weaknesses:**

(1) No classical operations‐research routing baselines are included; the framework uses one‐shot pre-trip routing, ignoring en-route replanning, incident response, and CAV platooning advantages.
(2) Results are reported as “best of three runs” without mean±CI or significance testing, and quantitative metrics for emissions or equity impacts are omitted.

**Strengths Contributions:**

(1) URB includes 29 real-world traffic networks, peak-demand data, a configurable SUMO simulation pipeline, JanuX route generation, and task/config templates to support end-to-end evaluation.
(2) The paper provides open‐source code for four MARL algorithms (IQL, IPPO, MAPPO, QMIX) and three baselines, complete with hyperparameters, random seeds, hardware usage, and a one‐click Code-Ocean capsule.
(3) Experiments show that most MARL methods struggle to outperform human routing, establishing a challenging baseline and clear direction for future improvements.4

---

> ### Author Rebuttal · Authors · 2025-07-31
>
> Thank you for your detailed assessment and valuable comments. We are delighted to see that URB's relevance, accessibility, and comprehensiveness, as well as our study's transparency, are identified and praised in your review. We have carefully considered your feedback and made the necessary adjustments to the manuscript. Our responses to each identified weakness are provided below, and we would be happy to provide further clarifications during the discussion period.
>
> ---
>
> > [W1a] No classical operations‐research routing baselines are included.
>
> We fully agree that the inclusion of OR-based baselines could potentially further challenge the ML-based solutions. However, our results indicate that the naive baselines sufficiently challenge the current SOTA; we have therefore omitted other classes of methods to focus on a more concise narrative. While this is the case for this particular study, we believe that progress towards addressing open research questions (lines 26-37) also involves adapting classical solutions to more dynamic and large-scale problem settings such as ours, which is also called for by prior work [1, 2]. Consequently, we have designed our benchmarking environment to be inclusive to accommodate various kinds of techniques. We are actively working on the implementations of a series of evolutionary algorithms (e.g., genetic algorithms) and other discrete optimization approaches from OR to enrich our leaderboard (they are already in the development cycle of our repository). We are hopeful that with further contributions in the coming iterations, URB will ultimately serve as a versatile testbed for domain-independent advancements in fleet routing tasks.
>
>
> > [W1b] The framework uses one‐shot pre-trip routing, ignoring en-route replanning, incident response, and CAV platooning advantages.
>
> Incorporating the phenomena you mentioned would certainly make the problem setting more comprehensive. However, we would like to clarify that our focus on one-shot pre-trip routing was intentional, as it provides an excellent starting point by isolating the key dynamics of human-AI interactions in a controlled setting. Notably, even this simplified task formulation has proven to be non-trivial, which, in our opinion, validates its suitability as an initial step. However, for the future directions, our benchmarking environment is designed to be extensible, and it contains provisions for the gradual introduction of such complexities.
>
> > [W2a] Results are reported as “best of three runs” without mean±CI or significance testing.
>
> The reason for reporting the best of three runs was largely due to high variance across repetitions (particularly evident in QMIX), which we hypothesize is caused by the increased non-stationarity and the large search space at our problem scale (e.g., 222 agents in Saint Arnoult, 1035 in Ingolstadt). Oftentimes, the mean was not meaningful, and we needed to resort to other metrics. To address the concerns about the statistical significance, we now extend our results with additional repetitions and confidence intervals. The changes will be reflected in our figures and numerical results, with the values reported below.
>
> **Table 1**: Scenario 1 results for three cities. Each value reports the mean and standard deviation across 5 runs with different seeds. $t_{pre}$ (same in each experiment for each network) and baseline results (deterministic, single-run only) are omitted for brevity.
>
> | Network (Algorithm) | $t^{\text{test}}$ | $t_{\text{CAV}}$ | $t_{\text{HDV}}$ | $c_{\text{all}}$ | $c_{\text{HDV}}$ | $c_{\text{CAV}}$ | $\Delta_{\text{V}}$ | $\Delta_{\text{L}}$ | **WR** |
> |---|:---:|:---:|:---:|:---:|:---:|:---:|:---:|:---:|:---:|
> | S.A. (IPPO) | 3.28 (0.004) | 3.33 (0.013) | 3.25 (0.008) | 0.63 (0.015) | 0.13 (0.004) | 1.38 (0.034) | -0.24 (0.067) | 0.06 (0.004) | 0.0 |
> | S.A. (IQL) | 3.36 (0.04) | 3.53 (0.104) | 3.24 (0.005) | 0.66 (0.0) | 0.14 (0.0) | 1.44 (0.004) | -0.37 (0.115) | 0.09 (0.021) | 0.0 |
> | S.A. (MAPPO) | 3.35 (0.049) | 3.51 (0.121) | 3.25 (0.004) | 0.66 (0.0) | 0.14 (0.004) | 1.45 (0.0) | -0.27 (0.129) | 0.09 (0.019) | 0.0 |
> | S.A. (QMIX) | 3.24 (0.08) | 3.21 (0.206) | 3.25 (0.004) | 0.65 (0.004) | 0.14 (0.005) | 1.43 (0.005) | -0.22 (0.034) | 0.03 (0.04) | 0.45 |
> | Provins (IPPO) | 2.9 (0.015) | 2.98 (0.04) | 2.85 (0.004) | 0.61 (0.271) | 0.31 (0.217) | 1.05 (0.356) | -0.52 (0.08) | 0.05 (0.009) | 0.0 |
> | Provins (IQL) | 2.91 (0.011) | 3.01 (0.027) | 2.85 (0.008) | 1.4 (0.104) | 0.92 (0.068) | 2.12 (0.183) | -0.58 (0.093) | 0.05 (0.007) | 0.0 |
> | Provins (MAPPO) | 2.93 (0.011) | 3.05 (0.024) | 2.84 (0.005) | 1.29 (0.162) | 0.83 (0.11) | 2.0 (0.247) | -0.69 (0.038) | 0.06 (0.004) | 0.0 |
> | Provins (QMIX) | 2.96 (0.005) | 3.14 (0.0) | 2.85 (0.0) | 0.85 (0.215) | 0.52 (0.176) | 1.35 (0.278) | -0.82 (0.033) | 0.08 (0.0) | 0.0 |
> | Ingolstadt (IPPO) | 4.41 (0.005) | 4.71 (0.03) | 4.21 (0.023) | 2.42 (0.497) | 1.9 (0.505) | 3.19 (0.495) | -0.52 (0.095) | 0.06 (0.004) | 0.0 |
> | Ingolstadt (IQL) | 4.46 (0.009) | 4.81 (0.024) | 4.23 (0.009) | 2.54 (0.546) | 1.93 (0.533) | 3.44 (0.562) | -0.69 (0.067) | 0.07 (0.0) | 0.0 |
> | Ingolstadt (MAPPO) | 4.45 (0.011) | 4.82 (0.019) | 4.21 (0.008) | 2.76 (0.599) | 2.16 (0.622) | 3.66 (0.562) | -0.72 (0.066) | 0.07 (0.004) | 0.0 |
> | Ingolstadt (QMIX) | 4.5 (0.14) | 4.87 (0.325) | 4.24 (0.015) | 1.83 (0.749) | 1.27 (0.71) | 2.67 (0.81) | -0.97 (0.235) | 0.06 (0.045) | 0.0 |
>
>
> > [W2b] Quantitative metrics for emissions or equity impacts are omitted.
>
> We refrained from reporting extra metrics like emissions or equity as the models were not calibrated and tested on those metrics. Such results can lead to misinterpretations by non-technical readers or policymakers. At this stage, we cannot guarantee that the SUMO-simulated impact of MARL-based CAV routing on emissions or equity is accurate. However, if one uses a model calibrated for that purpose in an external study, this can be documented and included in the benchmark. We decided to report this as a functionality to be exploited in future extensions, rather than reporting unverified and potentially misleading values. We reflect on this in the text now.
>
> ---
> [1] Ammouriova, M., Herrera, E. M., Neroni, M., Juan, A. A., Faulin, J. (2022). Solving vehicle routing problems under uncertainty and in dynamic scenarios: From simheuristics to agile optimization. Applied Sciences, 13(1), 101.
>
> [2] Psaraftis, H. N. (1995). Dynamic vehicle routing: Status and prospects. Annals of operations research, 61(1), 143-164.

---

### Official Review · Reviewer_3LBj · 2025-06-29

**Rating:** 5
**Confidence:** 3

**Summary:**

This paper introduces a new benchmark framework for evaluating the mixed human–CAV traffic routing on real city networks, author prepared 28 Île-de-France subregions via a synthetic demand pipeline and the Ingolstadt OD dataset with realistic trip patterns, it employs a rule-based Human Learning Model to stabilize human driver behavior, then “mutates” a share of vehicles into CAVs, and later trains those CAVs via MARL algorithms. The author compared baseline methods on metrics such as travel time, network speed/mileage change, training cost, and CAV win‐rate.

**Additional Feedback:**

1. There are some typos in the paper, for example, in the abstract, algorithmsrarely - typo: algorithms rarely


2. Glad to see the authors conclude with a bold result that the RL policies fail to outperform the humans; however, looking into the details, all RL methods record a 0 % “CAV wins” rate except QMIX in the smallest network. This uniform zero suggests either that the policies never actually update from random initialization or that the reward signal is not being backpropagated correctly. Based on the training curves in the Appendix, such as Scenario 2, Saint Arnoult, both MAPPO and IQL seem not to be converging, nor even learning. This raises a major concern about the algorithm's implementation correctness.


3. The paper introduces two sets of indices, τ for “today” and j for “earlier days,” but it does not mention how far back j goes. Is it all past days or only the most recent one? And I assume the j  = 0, 1, .... τ-1? If that is aligned with my understanding, I think it is better to explicitly define the j's range.


4. In the explanation of the equations, for example, in line 759, there are ε_i, εi,k and εi,k, τ, first, in the writing, the ε_i is in different format as εi,k and εi,k, τ, besides, in the remarks, author had set the values to zeros for ε and \gamma_{C} (Eq-1) and \gamma_{u} (Eq 3), would this be an oversimplification? It makes the human‐learning model very “clean,” but at the cost of realism.

**Dataset Code Accessibility:**

Yes

**Dataset Code Comments:**

The dataset is publicly available, and the code is also available on the open-sourced repository. The reproducibility is not a concern, yet I doubt the correctness of the implementation.

**Ethical Considerations:**

No, there are no or only very minor ethics concerns

**Final Justification:**

Please add the results in the rebuttal where the authors validate the hypotheses of RL performing badly into the paper.

**Limitations Weaknesses:**

1. The writing and presentation of the paper could be further improved. There are some typos, and the table results suggest reporting the mean and variance since the differences among various methods are very small.


2. The paper’s major concern is the correctness of the implementation for the RL methods, because it turns out that most of the RL methods’ performance are very bad, and reflected from the training curves, if the gradient is decreasing is hard to tell, e.g., MAPPO, IQL, QMI (line 817 - 819). This brings doubts about the usefulness of the work.


3. The paper at the beginning claimed four major directions with more than seven sub-questions on the RL for EAV systems (lines 26-37); however, due to the concern of the W2, this work might not be convincing enough to answer these questions.


4. It seems that the authors oversimplified the human-learning model by assigning zeros to the parameters in line 762.  This makes the model very “clean,” but at the cost of realism.

**Strengths Contributions:**

1. Addresses a critical real‐world gap: The paper clearly recognizes that, despite rapid advances in connected autonomous vehicles (CAVs), there is no standardized, large‐scale benchmark to evaluate CAV routing algorithms in realistic mixed‐traffic settings; by proposing URB, the authors provide a much‐needed platform that directly supports the development and comparison of CAV systems for actual urban deployment.


2. Provides a formal evaluation protocol: In the URB’s pipeline, it first training a human driver model to stability, then “mutating” a fixed fraction of agents into CAVs, and finally training and testing CAV policies under identical background conditions, this ensures experiment follows the same sequence and settings, which makes cross‐algorithm comparisons fair and repeatable.


3. High‐fidelity, real‐world demand scenarios: By incorporating origin–destination and departure‐time patterns drawn from national travel surveys for 28 Île‐de‐France regions and the RESCO InTAS dataset for Ingolstadt, URB tests routing algorithms against traffic demands that mirror true urban mobility; this realism boosts confidence that performance gains in the benchmark will translate to real‐city environments.

---

> ### Author Rebuttal · Authors · 2025-07-31
>
> Thank you very much for your detailed assessment and valuable feedback. We are grateful for your recognition of our framework's novelty, relevance, and realism, as well as your assessment of the real-world applicability. We have carefully considered your comments and have taken steps to address the weaknesses identified in your review. Our responses to your separate points are provided below, and we would be happy to offer further clarifications during the discussion period.
>
> ---
> > **[W1]** Improvements in the writing and presentation
>
> Thank you for highlighting this. We proofread the text and improved the clarity. We also enhanced our presentation in Table 1 by reporting the mean and standard deviations for each metric, also incorporating 2 newly added experiment runs, with the values provided below.
>
> **Table 1**: Scenario 1 results for three cities. Each value reports the mean and standard deviation across 5 runs with different seeds. $t_{pre}$ (same in each experiment for each network) and baseline results (deterministic, single-run only) are omitted for brevity.
>
> | Network (Algorithm) | $t^{\text{test}}$ | $t_{\text{CAV}}$ | $t_{\text{HDV}}$ | $c_{\text{all}}$ | $c_{\text{HDV}}$ | $c_{\text{CAV}}$ | $\Delta_{\text{V}}$ | $\Delta_{\text{L}}$ | **WR** |
> |---|:---:|:---:|:---:|:---:|:---:|:---:|:---:|:---:|:---:|
> | S.A. (IPPO) | 3.28 (0.004) | 3.33 (0.013) | 3.25 (0.008) | 0.63 (0.015) | 0.13 (0.004) | 1.38 (0.034) | -0.24 (0.067) | 0.06 (0.004) | 0.0 |
> | S.A. (IQL) | 3.36 (0.04) | 3.53 (0.104) | 3.24 (0.005) | 0.66 (0.0) | 0.14 (0.0) | 1.44 (0.004) | -0.37 (0.115) | 0.09 (0.021) | 0.0 |
> | S.A. (MAPPO) | 3.35 (0.049) | 3.51 (0.121) | 3.25 (0.004) | 0.66 (0.0) | 0.14 (0.004) | 1.45 (0.0) | -0.27 (0.129) | 0.09 (0.019) | 0.0 |
> | S.A. (QMIX) | 3.24 (0.08) | 3.21 (0.206) | 3.25 (0.004) | 0.65 (0.004) | 0.14 (0.005) | 1.43 (0.005) | -0.22 (0.034) | 0.03 (0.04) | 0.45 |
> | Provins (IPPO) | 2.9 (0.015) | 2.98 (0.04) | 2.85 (0.004) | 0.61 (0.271) | 0.31 (0.217) | 1.05 (0.356) | -0.52 (0.08) | 0.05 (0.009) | 0.0 |
> | Provins (IQL) | 2.91 (0.011) | 3.01 (0.027) | 2.85 (0.008) | 1.4 (0.104) | 0.92 (0.068) | 2.12 (0.183) | -0.58 (0.093) | 0.05 (0.007) | 0.0 |
> | Provins (MAPPO) | 2.93 (0.011) | 3.05 (0.024) | 2.84 (0.005) | 1.29 (0.162) | 0.83 (0.11) | 2.0 (0.247) | -0.69 (0.038) | 0.06 (0.004) | 0.0 |
> | Provins (QMIX) | 2.96 (0.005) | 3.14 (0.0) | 2.85 (0.0) | 0.85 (0.215) | 0.52 (0.176) | 1.35 (0.278) | -0.82 (0.033) | 0.08 (0.0) | 0.0 |
> | Ingolstadt (IPPO) | 4.41 (0.005) | 4.71 (0.03) | 4.21 (0.023) | 2.42 (0.497) | 1.9 (0.505) | 3.19 (0.495) | -0.52 (0.095) | 0.06 (0.004) | 0.0 |
> | Ingolstadt (IQL) | 4.46 (0.009) | 4.81 (0.024) | 4.23 (0.009) | 2.54 (0.546) | 1.93 (0.533) | 3.44 (0.562) | -0.69 (0.067) | 0.07 (0.0) | 0.0 |
> | Ingolstadt (MAPPO) | 4.45 (0.011) | 4.82 (0.019) | 4.21 (0.008) | 2.76 (0.599) | 2.16 (0.622) | 3.66 (0.562) | -0.72 (0.066) | 0.07 (0.004) | 0.0 |
> | Ingolstadt (QMIX) | 4.5 (0.14) | 4.87 (0.325) | 4.24 (0.015) | 1.83 (0.749) | 1.27 (0.71) | 2.67 (0.81) | -0.97 (0.235) | 0.06 (0.045) | 0.0 |
>
>
> > **[W2]** Correctness of the implementation for the RL methods
>
> Thank you for raising this concern. To conduct a fair and rigorous benchmarking study, we use direct adaptations of TorchRL's general-use and community-maintained algorithm implementations in our experiments. Our adaptation involves minimal to no changes in the training pipeline, and we verified correctness by analyzing the data flow across components. Each experiment script in URB comes with a link to the original TorchRL implementation for reference.
>
> We believe that the underlying reason goes beyond the correctness of the implementation and relates more to fundamental challenges of the problem at hand, such as increased non-stationarity, local observations, and credit assignment difficulty in large agent groups. Our problem formulation, supported by a diverse set of tasks, is intended to foster community contributions and position URB as an inclusive assessment ground for methodological advancements.
>
> In addition to encouraging community contributions, addressing the open research questions outlined in our paper also necessitates continuous effort and incremental extensions from our side. Accordingly, our further analysis allowed us to verify several hypotheses regarding the poor performance of RL methods tested so far. We found that eliminating temporal difference mechanisms (as our problem is single-step from each agent's perspective) and disabling experience sharing (effective in many settings [1,2], but disabling it boosts individual learning efficiency) improves the algorithm performance by a notable margin compared to generic implementations. Our preliminary results show improved convergence properties by surpassing our naive baselines with greater consistency. This extension is already in the development cycle of our repository, and the new results will be incorporated into our leaderboard and can be shared with the reviewers for further discussion, if needed.
>
> > **[W3]** Added potential for addressing open research questions
>
> The algorithmic challenges (identified in [W2]) evident in our results indeed show that reaching the far-reaching objectives will not be trivial. This reinforces the need for a community benchmark, on which the convincing answers to the stated questions can be made through a series of structured experiments. The desired objectives are achievable with the community contributions, and we believe that our benchmarking study challenges the community to progress towards the prescribed objectives, and URB provides a foundation for these advancements. We revised the corresponding section to clarify our intent by including your valuable point.
>
> > **[W4]** Human model parameterization
>
> That is true, adding new sources of non-determinism and heterogeneity into the environment makes the task even more challenging. We, despite implementing a complete and diverse human learning model, decided to start with a simplified deterministic and possibly sub-optimal behavioural model. We intend to include more realistic human models in the following tasks of URB when the SOTA algorithms consistently outperform humans on the simplified cases.
>
> > **[AF1]** Typos
>
> Thank you for pointing this out. We will proofread the text and carefully address all the typos and any other remaining editorial issues in the manuscript.
>
> > **[AF2]** Concerns about algorithm implementations
>
> We devoted significant attention to conducting a fair and robust benchmarking study, which provided us with the confidence to report such surprising yet discussion-worthy results. However, such bold results require additional care in interpretation. Our presentation (Sec. 4 and App. F.3) is centered around the quality of the solution found by the MARL algorithms, quantified by the average travel times. This, unfortunately, is not fully representative of the learning efficiency. Even though sometimes it reflects the policy improvements (like in line 824), occassionaly we observe unstable fluctuations in travel times (line 827) for the most part of the training process, even if a good solution is eventually found. The additional experiment statistics provided in our data repository (shared in Appendix B) contain further insights about the learning performance. These additional statistics, which can be further explored through additional analyses, indicate that our training pipeline effectively results in evolving AV route choice trends and loss minimization, which can provide a better context regarding the learning process.
>
> Furthermore, as we mentioned in our response to [W2], our choice of algorithm implementations is made to ensure alignment with community standards. One of our motivations for selecting TorchRL is that it operates with reusable and drop-in components, which allows different algorithm implementations to follow similar training pipelines, mostly differing by used components. For instance, our IPPO implementation differs from MAPPO only by line #279, and QMIX and VDN only differ by the used mixing module (VDNMixer vs. QMixer). This minimizes the risk of algorithm-specific implementation bias.
>
> > **[AF3]** Range of $j$ in Appendix D.3
>
> Thank you for noticing this ambiguity. While we do not explicitly state the range of the index $j$, the summation in Equation 1, $\dots \sum_{j=0}^{\tau-1} \alpha_j \dots$, is intended to indicate the bounds of $j$. We agree that this was not sufficiently clear, hence we revised the section to make it more explicit.
>
> > **[AF4]** Notation inconsistency in Appendix D.3 and realism in human learning model parameterization
>
> Thank you for raising this point. We revised our notation consistency. Our human model parameterization is indeed a deliberate simplification of the real-world heterogeneity in human decision-making. By removing unnecessary variance, we analyze the group-level interactions under controlled conditions. Although a less sparse configuration would be useful to assess a solution's practical performance, we believe that the matter of scalability will follow the identification of solutions effective in more simplistic settings. When the current algorithmic challenges (as also identified in [W2]) are addressed by future developments, URB will accommodate more realistic problem extensions (including different human model parameterizations) from day zero, thanks to its modular and flexible configuration structure.
>
> ---
> [1] Christianos, F., Schäfer, L., & Albrecht, S. (2020). Shared experience actor-critic for multi-agent reinforcement learning. Advances in neural information processing systems, 33, 10707-10717.
>
> [2] Gerstgrasser, M., Danino, T., & Keren, S. (2023). Selectively sharing experiences improves multi-agent reinforcement learning. Advances in Neural Information Processing Systems, 36, 59543-59565.

---

> > ### Comment · Reviewer_3LBj · 2025-08-05
> >
> > Thanks for the rebuttal. My concern about the bad performance of RL methods is not solved. I appreciate the authors ' explanation on possible alternative hypotheses to the performance, such as increased non-stationarity, local observations, and credit assignment difficulty in large agent groups - it could be better if the authors validate these hypotheses with results.

---

> > > ### Author Response · Authors · 2025-08-06
> > >
> > > We appreciate the reviewer's suggestion to validate our hypotheses with empirical results. We have conducted additional experiments to support our hypotheses regarding the poor RL performance (see Table 2 below) and report the results of improved RL-based approaches that provide better learning performance (see Table 3 below).
> > >
> > > ### A) Verifying hypotheses on the reasons for poor RL performance
> > >
> > > We conducted experiments on the St. Arnoult network, on one hand with halving and doubling the demand (number of agents) and using global observations (city-wide route selection history) on the other.
> > >
> > > Our results (in Table 2) show that the algorithmic performance decreases with the demand level and restriction of global information, as reflected in $\Delta_{\text{\%}}t^{\text{pre}}$ values. This suggests that locality of observations, congestion (source of non-stationarity), and size of the coordinating agent group (difficulty in credit assignment) correlate with algorithm performance in our setting. In URB tasks, these complexities (realism restrictions in access to global information, fleet sizes on the order of hundreds) are inevitable and yet to be addressed by methodological advancements.
> > >
> > > **Table 2:** Additional experiments conducted in St. Arnoult with varying trip demands and observations, using IQL and MAPPO algorithms. $t^{\text{pre}}$ and $t_{\text{CAV}}$ are the mean travel times of humans (before CAVs) and CAVs (after 1200 training episodes), respectively. $\Delta_{\text{\%}}t^{\text{pre}}$  is the improvement in $t_{\text{CAV}}$ compared to $t^{\text{pre}}$.
> > > | Experiment | $t^{\text{pre}}$ | $t_{\text{CAV}}$ | $\Delta_{\text{\%}}t^{\text{pre}}$ |
> > > |---|:---:|:---:|:---:|
> > > | IQL (Half demand) | 3.27 | 3.45 | -5.22% |
> > > | IQL (Original) | 3.15 | 3.53 | -10.76% |
> > > | IQL (Double demand) | 3.24 | 5.81 | -44.23% |
> > > | IQL (Global observations) | 3.15 | 3.26 | -3.37%|
> > > | MAPPO (Half demand)| 3.27 | 3.44 | -4.94% |
> > > | MAPPO (Original)| 3.15 | 3.45 | -8.70% |
> > > | MAPPO (Double demand) | 3.24 | 5.23 | -38.05% |
> > >
> > > ### B) Improving the RL-based methods
> > >
> > > This finding allowed us to develop alternative approaches that improve the learning performance. We were reluctant to share new results in the first rebuttal (partly due to strong policy on no sharing links and images within this track). We improved implementations of IQL and IPPO, which now beat the benchmarks with greater consistency and are good candidates to dominate the leaderboard and, hopefully, lead to the development of SOTA methods. This is an argument suggesting that the reason behind poor performance could be more complex than the implementation error. Specifically, we eliminated temporal credit assignment mechanisms (bootstrapping term in DQN loss, and critic estimation in PPO advantage) and used only local experiences in learning. Both implementations improve the performance of their predecessors, as reported in Table 3, with 100% WR in S.A and Provins by IQL, indicating that the CAVs consistently achieved shorter mean travel times than human drivers after training. We are updating the repository with improved methods and results (in the development branch).
> > >
> > > **Table 3**: **Effect of improving the IPPO and IQL implementations.** Results of the independent learning algorithms, their improved implementations (\*), and naive baselines in Scenario 1. Each value reports the mean and standard deviation across 5 seeded runs, except for the baselines. $t_{\text{CAV}}$ is the mean travel time of the CAVs, $c_{\text{CAV}}$ is the cost of learning for CAVs (as introduced in Sec. 3), and the winrate **WR** is the percentage of experiment runs where $t_{\text{CAV}}<t^{\text{pre}}$.
> > >
> > > | Network (Algorithm) | $t_{\text{CAV}}$ | $c_{\text{CAV}}$ | **WR** |
> > > |---|:---:|:---:|:---:|
> > > | S.A. (IPPO*) | 3.31 (0.05) | 0.8 (0.06) | 0.0 |
> > > | S.A. (IPPO) | 3.33 (0.013) | 1.38 (0.034) | 0.0 |
> > > | S.A. (IQL*) | 3.02 (0.011) | 1.41 (0.0) | 1.0 |
> > > | S.A. (IQL) |3.53 (0.104) | 1.44 (0.004) | 0.0 |
> > > | S.A. (AON) | 3.01 | 1.21 | 1.0 |
> > > | S.A. (Random) | 3.58 | 1.36 | 0.0 |
> > > | Provins (IPPO*)| 2.88 (0.013) | 0.64 (0.09) | 0.0 |
> > > | Provins (IPPO)| 2.98 (0.04) | 1.05 (0.356) | 0.0 |
> > > | Provins (IQL*)| 2.76 (0.008) | 1.19 (0.186) | 1.0 |
> > > | Provins (IQL)| 3.01 (0.027) | 2.12 (0.183) | 0.0 |
> > > | Provins (AON) | 2.76 | 0.99 | 1.0 |
> > > | Provins (Random) | 3.04 | 0.95 | 1.0 |
> > > | Ingolstadt (IPPO*)| 4.56 (0.025) | 1.86 (0.535) | 0.0 |
> > > | Ingolstadt (IPPO)| 4.71 (0.03) | 3.19 (0.495) | 0.0 |
> > > | Ingolstadt (IQL*)| 4.22 (0.015) | 2.01 (0.27) | 0.0 |
> > > | Ingolstadt (IQL) | 4.81 (0.024) | 3.44 (0.562) | 0.0 |
> > > | Ingolstadt (AON) | 4.37 | 0.24 | 0.0 |
> > > | Ingolstadt (Random) | 4.81 | 1.74 | 0 |
> > >
> > > These new results, in our opinion, not only demonstrate the challenges of the fleet urban routing problems but also prove the usefulness of URB in the development of effective methods. The performance improvements show that our research questions can be effectively addressed by methodological advancements and experimental studies.

---

> > > > ### Comment · Reviewer_3LBj · 2025-08-06
> > > >
> > > > Thanks for the newly added results. Now I’m convinced that this is a challenging benchmark with merits for RL methods to solve. Will update my score.

---

### Official Review · Reviewer_g2QS · 2025-07-02

**Rating:** 5
**Confidence:** 3

**Summary:**

This paper presents a benchmark for urban route planning with the inclusion of connected autonomous cars, particularly for evaluating reinforcement learning algorithms for optimization. Traffic demands are generated for 29 real-world traffic networks using current methods for traffic synthesis combined with population statistics from the simulated regions. These networs and traffic demands form a corpus of tasks on which to evaluate optimization algorithms. Four MARL algorithms are evaluated and compared to three baseline models. Results on the benchmark suggest that current approaches may not improve over the baselines, indicating a need for further study.

**Additional Feedback:**

A space is missing on the abstract on line 13. "algorithmsrarely"

**Dataset Code Accessibility:**

Yes

**Dataset Code Comments:**

Dataset availability has followed the NeurIPS guidelines and code is readily available via GitHub.

**Ethical Considerations:**

No, there are no or only very minor ethics concerns

**Final Justification:**

The benchmark improves over prior settings with the inclusion of CAVs, learnable driving behaviors, and additional traffic scenarios.

**Limitations Weaknesses:**

As identified in section 5, demand patterns remain fixed across experiments, both in origin-destination pairs and departure times. While I believe this to be a significant simplification, the lack in effectiveness of the evaluated algorithms demonstrates that the simplified problem still adds value to evaluating algorithms. It may even be beneficial to have this simplification in the course of development of improved methods as it isolates the problem from a complex source of nonstationarity.

Given the basis on the cited work [2], it would be helpful to more explicitly define what is contributed with this paper, which was not present in the prior. Certainly, the datasets are contributed. As stated in section 1.1, no other single benchmark combines all aspects. It might be beneficial to chart these in a table which indicates which aspects are lacking from other benchmarks.

**Strengths Contributions:**

The benchmark contributes a reasonably diverse dataset for the target task which could enable more accurate evaluations. The contributed code provides a well-documented and easy to use framework for simulating the dataset. Further the writing is easy to follow and clear.

---

> ### Author Rebuttal · Authors · 2025-07-31
>
> Thank you for your thoughtful evaluation and constructive remarks. We appreciate your recognition of the strengths of our work, particularly in the areas of our dataset, codebase, documentation, and presentation in the paper. After careful consideration, we provide the necessary clarifications below and address the points you raised for improvements.
>
> ---
>
> > **[W1]** As identified in section 5, demand patterns remain fixed across experiments, both in origin-destination pairs and departure times. While I believe this to be a significant simplification, the lack in effectiveness of the evaluated algorithms demonstrates that the simplified problem still adds value to evaluating algorithms. It may even be beneficial to have this simplification in the course of development of improved methods as it isolates the problem from a complex source of nonstationarity.
>
> We fully agree that the setting is far from a digital twin replication of real-world urban mobility. In our simulation framework, RouteRL [1], we allow for diverse settings, bringing the experiments closer to reality, yet at the cost of computation time. As you have pointed out, a simplified problem setting allows us to analyze the disturbances purely caused by the coexistence of humans and automated (possibly coordinated) decision-making systems in a mixed traffic system, isolated from undesirable effects introduced in a more complicated and fine-grained experimental setting. Furthermore, our ultimate objective is to pave the way for more elaborate empirical analyses, and with URB, we contribute a relatively simplified (yet seemingly challenging) starting point for this line of research. Our framework is designed to accommodate problem extensions, which hopefully will be addressed by the community as soon as the fundamental issues are solved and basic knowledge is gathered on the simpler tasks introduced here.
>
> > **[W2]** Given the basis on the cited work [2], it would be helpful to more explicitly define what is contributed with this paper, which was not present in the prior. Certainly, the datasets are contributed. As stated in section 1.1, no other single benchmark combines all aspects. It might be beneficial to chart these in a table which indicates which aspects are lacking from other benchmarks.
>
> The simulation framework of RouteRL [1] provides a programming interface for defining and simulating a variety of AV deployment scenarios on microscopic scale, but its usage is limited to user-defined tasks and implementations. URB provides a full-fledged benchmarking environment, which is much needed yet lacking in our line of research. The framework contains a reusable and reproducible experimental suite, paired with an extensive collection of real-world-inspired tasks, without requiring users to engage with RouteRL's internal workflow and API usage. Moreover, as stated in the section you pointed out, our problem involves additional complexities that constitute a novel difficulty for MARL algorithms in contrast to standardized benchmarks. Following your suggestion, we make this novelty more concrete by adding to our paper the following table comparing different aspects of URB with other benchmarks established in the scientific literature. We also generally improve our presentation to clarify the added value of our contribution for the camera-ready version.
>
>
> **Table 1:** A comparison of URB with comparable and prominent MARL benchmarks. Only URB uses diverse, realistic traffic networks as an environment. [2] contains mainly grid world or multi-robotic simulations and has mostly intersecting evaluation as [3] has. [4] was designed to benchmark traffic signal control using both realistic (parts of a city) and toy (like 4x4 Grid) networks. Only one environment in [5] is a realistic network; however, it has a very simple architecture.
>
> | **Benchmark and its objective** | **Environment Diversity** | **Coverage** | **Possible Extensions** |
> |---------------|---------------------------|----------------|----------------------------|
> | **URB** (ours) - Analysis of CAV impact on large-scale urban routing optimization| Scenario types; network sizes; AV behaviors; up to 6924  agents | 7 algorithm implementations; 29 networks                        | Extendable networks & scenario catalog; varying agent count; custom algorithm implementations |
> | **BenchMARL** [2] - Enabling standardized benchmarking across algorithms, models, and environments| Task variety; reward function types                | 9 algorithm implementations; 5 environments; 2–49 tasks | Extendable algorithm, model, and task catalog                   |
> | **EPyMARL** [3] - Comparing MARL approaches in environments of different difficulty | Observability settings; difficulty variety; 2–10 agents | 9 algorithm implementations; 5 environments| –|
> | **RESCO** [4] - Comparison of traffic signal controllers | Real-world city network segments | 8 algorithm implementations; 8 networks| –|
> | **FLOW** [5] - Learning control laws for autonomous vehicles | 14–22 agents| 6 networks| - |
>
>
> > **[AF1]** A space is missing on the abstract on line 13. "algorithmsrarely"
>
> Thank you for pointing out this oversight. We have now fixed it for the camera-ready version.
>
> ---
> [1] Akman, A. O., Psarou, A., Gorczyca, Ł., Varga, Z. G., Jamróz, G., & Kucharski, R. (2025). RouteRL: Multi-agent reinforcement learning framework for urban route choice with autonomous vehicles. arXiv preprint arXiv:2502.20065.
>
> [2] Bettini M., Prorok A., Moens V. (2024). BenchMARL: Benchmarking Multi-Agent Reinforcement Learning. Journal of Machine Learning Research 25 (2024), no. 217 pp. 1--10.
>
> [3] Papoudakis G., Christianos F., Schäfer L., Albrecht S. V. Benchmarking Multi-Agent Deep Reinforcement Learning Algorithms in Cooperative Tasks. Proceedings of the Neural Information Processing Systems 2021 Datasets and Benchmarks Track.
>
> [4] Ault J., Sharon G. Reinforcement Learning Benchmarks for Traffic Signal Control. Proceedings of the Neural Information Processing Systems 2021 Datasets and Benchmarks Track.
>
> [5] C. Wu, A. R. Kreidieh, K. Parvate, E. Vinitsky, A. M. Bayen. Flow: A Modular Learning Framework for Mixed Autonomy Traffic. IEEE Transactions on Robotics (2022) vol. 38, no. 2, pp. 1270-1286.

---

> > ### Comment · Reviewer_g2QS · 2025-08-07
> >
> > Thank you for your responses and proposed addition. I do not hold the same concern over algorithm correctness as other reviewers as the failure of RL algorithms on new benchmarks is frequently reported. I believe the contribution here is sufficient for acceptance and therefore raise my score.

---

### Note · Authors · 2025-08-12

We appreciate the reviewers constructive engagement and their recognition of URB novelty, comprehensiveness, and potential to advance research in CAV routing and in MARL. The rebuttal allowed us to **refine the manuscript, improve presentation, and clarify contributions**.

A concern raised by multiple reviewers was the poor performance of RL algorithms relative to naive baselines. We addressed this comprehensively by:
* verifying correctness of MARL implementations against the original TorchRL,
* conducting controlled experiments that confirmed performance degradation relates to fundamental challenges: non-stationarity, partial observability, and credit assignment in large agent groups; rather than flawed implementations
*  introducing improved IPPO and IQL variants which now consistently outperform human and naive baselines (100% win rate in smaller networks).

Consequently, Reviewer *3LBj* explicitly acknowledged that the new results convinced them of the benchmark’s merit, and subsequently updated (raised?) their score, while reviewer *g2QS* _now believe the contribution here is sufficient for acceptance and therefore raise my score_.

We also addressed other, smaller issues, like: presentation and statistical rigour by extending evaluations to five seeds, reporting mean +/- std, improving notation, clarifying index ranges, and adding a comparative table showing URB’s unique combination of realism, diversity, and reproducibility. Simplifications in the current setting (fixed demand, one-shot pre-trip routing, simplified human model) remain intentional to isolate algorithmic performance, with URB’s modular design enabling future extensions to dynamic traffic, incident response, en-route replanning, and richer human behaviour models. While classical OR baselines were omitted for narrative focus, we are implementing them (and evolutionary methods) for future leaderboard releases. Functionality for emissions and equity metrics will be available once calibrated models are used.

In summary, after discussion URB now, hopefully, delivers:
1.  the first standardized, large-scale MARL benchmark for urban CAV routing on real networks;
2. reproducible baselines and protocols for fair cross-method comparison; and
3.  empirical and now-validated evidence that realistic-scale MARL presents fundamental challenges.

*We believe URB will catalyze methodological advances and serve as a long-term community resource.*

---

### Decision · Program_Chairs · 2025-09-18

**Decision:**

Accept (poster)

**Comment:**

**Summary:**

This work proposes URB (Urban Routing Benchmark), an open-source framework for evaluating routing strategies in mixed human–CAV traffic. It integrates 29 real-world traffic networks, peak-hour demand data, and a simulation pipeline that models human drivers and allows a subset of vehicles to be mutated into CAVs. URB provides reproducible tasks, templates, and implementations of four MARL algorithms alongside three baselines. This work has experiments across multiple cities that reveal that current MARL methods rarely outperform simple baselines or human routing. The authors claim this demonstrates both the need for more effective algorithms and the value of URB as a standardized benchmark.

**Strengths:**

- URB integrates 29 real-world urban networks with high-fidelity demand data, providing a scalable and realistic environment for CAV routing research.

-  The framework ensures fair comparisons across algorithms by modeling human drivers, mutating CAVs, and supplying open-source implementations, task templates, and reproducibility resources.

- Systematic experiments reveal that state-of-the-art MARL methods struggle to surpass human or baseline routing, establishing a challenging benchmark and motivating future algorithmic advances.

**Weaknesses and rebuttal discussions:**

- The current MARL methods rarely surpass simple baselines, though this highlights the challenge rather than a flaw of the benchmark.

- The study uses simplified settings such as one-shot routing and stable human drivers; however, the authors clarified these were deliberate choices for clarity and extensibility.

- Initial concerns about missing OR baselines, limited metrics, and lack of statistical reporting were addressed in the rebuttal with clarifications and additional results.


**Recommendation: Accept.** The paper makes a timely contribution by introducing URB, a large-scale, open-source benchmark for CAV routing in mixed human–CAV traffic. It provides realistic datasets, reproducible pipelines, and standardized evaluation protocols, which will serve as a valuable resource for the community. While the experiments show that current MARL methods underperform, this result underscores the benchmark’s importance and sets the foundation for future advances. The rebuttal addressed the reviewer's concerns.